# Bayesian age models and stacks: Combining age inferences from radiocarbon and benthic $\delta^{18}$O stratigraphic alignment

Taehee Lee[1], Devin Rand[2], Lorraine E. Lisiecki[2], Geoffrey Gebbie[3], Charles Lawrence[4]

[1]Department of Statistics, Harvard University, Cambridge, USA
[2]Department of Earth Science, University of California, Santa Barbara, 93106, USA
[3]Physical Oceanography Department, Woods Hole Oceanographic Institution, Woods Hole, 02543, USA
[4]Division of Applied Mathematics, Brown University, Providence, 02906, USA

Taehee Lee and Devin Rand contributed equally to this work.

*Correspondence to*: Devin Rand (drand@ucsb.edu), Taehee Lee (taehee_lee@fas.harvard.edu)

**Abstract.** Previously developed software packages that generate probabilistic age models for ocean sediment cores are designed to either interpolate between different age proxies at discrete depths (e.g., radiocarbon, tephra layers, or tie points) or perform a probabilistic stratigraphic alignment to a dated target (e.g., of benthic $\delta^{18}$O) and cannot combine age inferences from both techniques. Furthermore, many radiocarbon dating packages are not specifically designed for marine sediment cores and default settings may not accurately reflect the probability of sedimentation rate variability in the deep ocean, requiring subjective tuning of parameter settings. Here we present a new technique for generating Bayesian age models and stacks using ocean sediment core radiocarbon and probabilistic alignment of benthic $\delta^{18}$O data, implemented in a software package named BIGMACS (Bayesian Inference Gaussian Process regression and Multiproxy Alignment of Continuous Signals). BIGMACS constructs multiproxy age models by combining age inferences from both radiocarbon ages and probabilistic benthic $\delta^{18}$O stratigraphic alignment and constrains sedimentation rates using an empirically derived prior model based on 37 $^{14}$C-dated ocean sediment cores (Lin et al., 2014). BIGMACS also constructs continuous benthic $\delta^{18}$O stacks via a Gaussian process regression, which requires a smaller number of cores than previous stacking methods. This feature allows users to construct stacks for a region that shares a homogeneous deep water $\delta^{18}$O signal, while leveraging radiocarbon dates across multiple cores. Thus, BIGMACS efficiently generates local or regional stacks with smaller uncertainties in both age and $\delta^{18}$O than previously available techniques. We present two example regional benthic $\delta^{18}$O stacks and demonstrate that the multiproxy age models produced by BIGMACS are more precise than their single proxy counterparts.

## 1 Introduction

The accuracy with which ocean sediment core data can reconstruct the timing of past climate events depends on the quality of the core's age model (i.e., estimates of age as a function of core depth). However, age models are often constrained by only a single dating proxy type. A common technique is radiocarbon dating, which directly dates individual sediment layers.

However, this method is restricted to the last 55 ka BP, suffers from variable surface reservoir ages (Waelbroeck et al., 2001; Sikes et al., 2016; Stern & Lisiecki, 2013; Skinner et al., 2019), and radiocarbon data are often lower resolution than benthic $\delta^{18}O$ data. Radiocarbon age models are sometimes supplemented with stratigraphic tie points to a dated target, however this method requires the subjective identification of shared features that are often recorded in different archives. An alternative technique is the stratigraphic alignment of benthic $\delta^{18}O$ to a target stack (e.g., Imbrie et al., 1984; Lisiecki & Raymo, 2005),

which represents the mean benthic $\delta^{18}O$ signal across multiple cores. Benthic $\delta^{18}O$ is often measured at higher resolution than radiocarbon data, but this dating technique provides only relative age information between cores by assuming that the input and target have synchronous benthic $\delta^{18}O$ signals. Temporal offsets between the aligned records can cause age errors in the aligned age model  (Skinner & Shackleton, 2005; Labeyrie et al., 2005; Waelbroeck et al., 2011; Stern & Lisiecki, 2014; Lund et al., 2015).

Software packages exist to produce age models by interpolating between age proxies (such as radiocarbon ages, tephra layers, or/and tie points; Blaauw & Christen 2011; Lougheed & Obrochta, 2019), or by performing a probabilistic benthic $\delta^{18}O$ alignment (in which residuals between input and target records are minimized; Lin et al., 2014; Ahn et al., 2017), but none of these packages can probabilistically combine age inferences from both dating techniques. While one study presented a Bayesian multiproxy age model for a single core from the Arctic Ocean, the methodology is specific to the high latitude region

in which radiocarbon data is unreliable and aligned porosity rather than benthic $\delta^{18}O$ (Muschitiello et al., 2020). Furthermore, many age modelling software packages were not specifically designed for marine sediment cores (Ramsey, 1995; Haslett & Parnell, 2008; Blaauw, 2010; Blaauw & Christen 2011) and default settings may not accurately reflect the probability of sediment accumulation rate variability in marine settings. Users must often subjectively choose parameter settings which may ultimately affect the interpretation of paleoclimate records.

Here we present a new technique for generating Bayesian age models and stacks of ocean sediment core data, implemented in a software package named BIGMACS (Bayesian Inference Gaussian Process regression and Multiproxy Alignment of Continuous Signals). BIGMACS constructs radiocarbon age models, benthic $\delta^{18}O$ age models, and multiproxy age models which combine age inferences from both radiocarbon ages and $\delta^{18}O$ stratigraphic alignment. Radiocarbon ages directly date sediment layers while benthic $\delta^{18}O$ provides relative age constraints between radiocarbon ages and beyond 55 ka

BP. We use the term "multiproxy" to indicate the combined inference from two types of "age proxies": absolute age information provided by radiocarbon and relative age information from the stratigraphic alignment of benthic $\delta^{18}O$. Note that this method is distinct from an alignment of multiple climate proxies (e.g., benthic and planktonic $\delta^{18}O$). BIGMACS can also probabilistically incorporate other types of age information at specified depths, such as inferences from tephra layers, magnetic reversals, or user-identified tie points. Sedimentation rates are realistically constrained with an empirically derived prior model

from Lin et al. (2014) rather than subjective parameter settings. Median age models and their uncertainties are defined by the distribution of Markov Chain Monte Carlo (MCMC) samples. The distribution of MCMC samples at a given depth of a radiocarbon age model reflects the absolute age uncertainty of the sediment. However, $\delta^{18}O$ age model uncertainty reflects

only the relative age uncertainty and excludes the absolute age uncertainty of the alignment target. BIGMACS does not use any orbital tuning unless users choose to align to a target stack that has been orbitally tuned.

65        Another functionality of BIGMACS is the automated construction of multiproxy benthic $\delta^{18}O$ stacks using an iterative process that simultaneously considers the probabilistic fit to both absolute age information (e.g., from radiocarbon dates) and relative age information from alignment of all cores' benthic $\delta^{18}O$ signals. Age models for each core are constructed by aligning benthic $\delta^{18}O$ to the stack from the previous iteration, and then a new stack is calculated from the aligned $\delta^{18}O$ from every core. Radiocarbon ages (if included) help constrain the age models for their respective cores during each iteration of stack

construction. Similar to "errors-in-variables" regression, which is used to construct the Intcal20 curve due to uncertainty in both the radiocarbon measurements and their calendar ages (Reimer et al., 2020; Heaton et al., 2020), BIGMACS calculates a time series of mean and variance for benthic $\delta^{18}O$ by performing Gaussian process regressions (Rasmussen and Williams, 2006) across MCMC age model samples. The resulting stack variance is a combination of both age model uncertainty from individual cores and the spread of benthic $\delta^{18}O$ from every core. This method requires fewer cores than previous stacking

methods (e.g., Ahn et al., 2017; Lisiecki & Stern, 2016) and, thus, allows users to construct target stacks from a small number of neighbouring cores that share homogeneous $\delta^{18}O$ signals.

       Section 2 provides a summary of some common techniques used for radiocarbon dating, $\delta^{18}O$ alignment and $\delta^{18}O$ stack construction. Section 3 describes the statistical methods used in BIGMACS, including an overview of the Bayesian framework, the prior model that constrains sedimentation rates, and the likelihood models for different proxy types. We also

describe the methods used to draw MCMC age model samples and the regression technique employed to construct continuous stacks from a small number of cores. In section 4, we present two example regional Atlantic stacks: a Deep Northeast Atlantic (DNEA) stack, and an Intermediate Tropical West Atlantic (ITWA) stack. The two stacks are composed of 6 and 4 cores respectively, that are chosen based on an evaluation of their water mass histories. In section 5, we compare a multiproxy age model, a $\delta^{18}O$-only age model, and a radiocarbon-only age model for one additional core. We demonstrate that age model

precision is increased when using both radiocarbon ages and $\delta^{18}O$ alignment. Finally, we discuss potential future applications of BIGMACS and the factors affecting its runtime.

## 2 Background

### 2.1 Radiocarbon Age Models

       Radiocarbon ages must be calibrated from [14]C years to calendar years with a calibration curve that accounts for the

changing magnetic fields of the Sun and Earth, solar storms, and variations in the terrestrial carbon cycle (Reimer et al., 2020; Heaton et al., 2020; Heaton et al., 2021). The uncertainty of the calibrated age is a combination of the calibration curve uncertainty, the radiocarbon measurement uncertainty, the time-dependent local reservoir age offset from the calibration curve ($\Delta R$) and the associated reservoir age uncertainty. Techniques to calibrate radiocarbon ages have evolved from interpolation techniques such as Calib (Stuiver & Reimer, 1993) to Bayesian calibration methods (e.g., Oxcal by Ramsey, 1995; Bcal by

95 Buck and Christen, 1999; Matcal by Lougheed & Obrochta, 2016) which typically generate asymmetric, nonparametric calendar age distributions due to slope changes in the calibration curve.

   Planktonic foraminiferal radiocarbon dates must be corrected for the reservoir age of the surface ocean relative to the atmosphere or calibrated with a curve that accounts for the reservoir age of the surface ocean (e.g., the Marine20 curve; Heaton et al., 2020). Previous studies have used different methods to estimate past reservoir ages, including using modern

100 measurements from the Global Ocean Data Analysis Project (GLODAP, Key et al., 2004, Waelbroeck et al., 2019) and the Calib database (Reimer & Reimer, 2001), comparing stratigraphically aligned age models with radiocarbon age models (Stern & Lisiecki, 2013; Skinner et al., 2021), and modelled reservoir ages from a Large Scale Geostrophic Ocean General Circulation Model (LSG-OGCM, Butzin et al., 2020; Butzin et al., 2017, Langner & Mulitza 2019; Heaton et al., 2020).

   Constructing a sediment core age model, which estimates sediment ages for all core depths, from a sequence of

105 radiocarbon ages requires assumptions or models of the core's evolving sedimentation rate between dated intervals. The median age model and age model uncertainty depend on the radiocarbon calibration method, the applied sedimentation rate constraints, and the outlier identification procedure (Christen, 1994; Ramsey, 2009b, Christen & Peréz, 2009). Multiple software packages have been published to construct probabilistic radiocarbon age models that apply a variety of statistical techniques (e.g., Ramsey, 1995, 2001, 2008, 2013; Blaauw & Christen, 2005; Haslett & Parnell 2008; Blaauw, 2010; Blaauw

110 & Christen, 2011; Lougheed & Obrochta, 2019).

   Oxcal (Ramsey, 1995) provides modelling routines for multiple depositional environments; the routine known as the P_Sequence is commonly used for modelling marine and lacustrine cores. P_Sequence uses a Poisson process in which the number of depositional events per unit depth is determined by a tuneable, user-specified parameter which affects the uncertainty of the age model. Oxcal also includes multiple options to identify outliers, including an agreement index which

115 measures the overlap between the posterior distribution of the age model and the radiocarbon likelihood at depths where radiocarbon ages exist.

   Bchron (Haslett & Parnell, 2008) constructs age-depth models using a monotone Markov process and piecewise linear interpolation paths with random durations. Bchron requires few user-specified parameter settings and posits less prior knowledge on sedimentation rate constraints; thus, age models constructed with Bchron often have larger age uncertainties

120 than other software packages, particularly for radiocarbon records of low resolution (Blaauw & Christen, 2011). Bchron identifies two types of outliers based on the shift required to satisfy the monotonicity constraint. Standard outliers have a prior probability of 5% and require a shift defined *a priori* by a normal distribution with variance equal to double the radiocarbon analytical measurement error. Larger outliers have a prior probability of 0.1% and are excluded from the age model construction process.

125 Bacon (Blaauw & Christen, 2011) separates cores into fixed segments and uses an auto-regressive gamma process to simulate sedimentation rates. The user specifies tuneable priors for a beta distribution that controls age model autocorrelation and a gamma distribution that governs sedimentation rate variability. Radiocarbon ages are modelled with a generalized student's t-distribution (Christen & Peréz, 2009) that scales the error associated with radiocarbon measurements. The amount

of scaling depends on two parameters which are set by default to assign a 70% chance that the reported error was underestimated by a factor between 1 and 2. Christen & Peréz (2009) explain that the choice of these parameter values is a "practical guideline" which they estimated to reflect the state of radiocarbon data at the time.

Undatable (Lougheed & Obrochta, 2019) uses a Monte Carlo sampling algorithm designed to emulate statistical models of sedimentation rate variability with the goal of producing quick runtimes. Users set two parameters: a scaling parameter that scales age uncertainties at the midpoints between radiocarbon ages and a bootstrapping percent that provides a framework to address outlier radiocarbon ages. These parameters have large effects on the resulting age model, requiring the user to select appropriate values, e.g., according to recommendations in Lougheed & Obrochta, (2019), rather than relying on a prior model of sedimentation rate variability.

## 2.2 Benthic $\delta^{18}$O Age Models

In the calcite shells of foraminifera, the ratio of $^{18}$O to $^{16}$O measured relative to a standard, denoted $\delta^{18}$O, is a proxy for global ice volume, local water temperature and the local $\delta^{18}$O of seawater, which often correlates with salinity. Due to the relatively homogeneous temperature and salinity changes of the deep ocean, previous studies have assumed benthic $\delta^{18}$O changes synchronously (Shackleton, 1967) and have used the proxy as a global stratigraphic signal to construct ocean sediment core age models (e.g., Pisias et al., 1984; Lisiecki and Raymo, 2005). The most conservative technique for aligning records to a target is to assume that large, easily identifiable features in the signals, such as glacial terminations, occurred simultaneously, create tie points between these features, and linearly interpolate between the tie points (e.g., Huybers & Wunsch, 2004). However, this linear interpolation method may misalign smaller scale features due to changes in sedimentation rates between tie points.

Software packages have been published that automate the alignment process and optimize the fit of the entire signal. Lisiecki & Lisiecki (2002) developed the deterministic software package Match, which utilizes dynamic programming to minimize a cost function based on sedimentation rate changes and the sum-of-square error misfit between signals. Match was used to align 57 benthic $\delta^{18}$O records and construct the global "LR04" Plio-Pleistocene stack (Lisiecki & Raymo, 2005) and a 1.5-Myr multiproxy geomagnetic paleointensity and $\delta^{18}$O stack (Channell et al., 2009).

The Bayesian package HMM-Match (Lin et al., 2014) performs a point-based alignment using a hidden Markov model and returns estimates of alignment uncertainty based on the distribution of MCMC age model samples. HMM-Match considers the probability of every possible alignment given the fit to the alignment target and the modelled sedimentation accumulation rate changes. The probability of a given benthic $\delta^{18}$O residual to the target is modelled with a fixed Gaussian distribution based on the record's $\delta^{18}$O residuals and a mean shift from the target. Sedimentation rates are realistically constrained using a log-normal mixture distribution fit to normalized sedimentation rate estimates derived by linearly interpolating between calibrated radiocarbon ages in 37 cores.

Heaton et al., (2013) presents an age model construction method which uses a Gaussian process regression to interpolate between benthic $\delta^{18}$O tie points. The method incorporates uncertainty from the target age model, tie point

identification, and interpolation between tie points and was used to construct chronologies for records incorporated into the IntCal13 and Intcal20 curve (Reimer et al., 2013; Reimer et al., 2020). Heaton et al., (2013) argue against using a deterministic automated alignment process (e.g., Lisiecki and Lisiecki, 2002) due to a lack of uncertainty estimates and concerns about

aligning across different proxy types which may differ in sensitivity to climate responses. We assert that using BIGMACS to align across a set of sediment cores with homogeneous signals of the same proxy (such as benthic $\delta^{18}$O in neighbouring cores), addresses these concerns. BIGMACS formally incorporates multiple sources of age uncertainty to create probabilistic alignments that are both more informative and less subjective than tie point identification.

Diachronous benthic $\delta^{18}$O signals are an additional source of uncertainty in benthic $\delta^{18}$O aligned age models. Previous

studies have identified temporal offsets up to 4 kyr between $\delta^{18}$O records during terminations (Skinner & Shackleton, 2005; Lisiecki & Raymo, 2009; Stern & Lisiecki, 2014). Because stratigraphic alignment relies on the assumption that benthic $\delta^{18}$O between the input and the target core varies synchronously, these offsets can cause age errors in $\delta^{18}$O-aligned age models. Thus, without a direct dating proxy (e.g., radiocarbon, tephra, etc.), $\delta^{18}$O stratigraphic alignment is an inadequate tool to study the sequence of climate responses at different locations during glacial terminations (e.g., Khider et al., 2017) or millennial-

scale events. Causes of offsets in the timing of benthic $\delta^{18}$O change include asynchronous surface signals, changes in deep ocean water mass geometry, or/and different deep water transit times (Gebbie, 2012). To mitigate the impacts of diachronous $\delta^{18}$O change, benthic $\delta^{18}$O alignment should ideally be restricted to cores which have experienced a similar history of deep water mass change. We present one method to identify cores with synchronous benthic $\delta^{18}$O signals in section 4.1.

## 2.3 Benthic $\delta^{18}$O Stacks

Benthic $\delta^{18}$O stacks are used as a common framework by which new paleoceanographic measurements are compared and are often used as targets during stratigraphic alignment (e.g., Imbrie et al., 1984; Lisiecki & Raymo, 2005; Channell et al., 2009). Stacks require that the individual $\delta^{18}$O records are first aligned to have comparable relative or absolute ages so that each point in the stack represents a snapshot of $\delta^{18}$O values from multiple locations at the same time. Inaccuracy in relative age estimates between cores will typically decrease the signal-to-noise ratio of the stacked signal, but over-alignment of noise in

the signals could artificially enhance variability that was not globally synchronous. The risk of over-alignment can be reduced by placing constraints on sedimentation rate variability (e.g., Lisiecki & Lisiecki, 2002; Lin et al., 2014).

To create a stack using software that performs pairwise alignments of cores, all $\delta^{18}$O records to be included in the stack are aligned to a single target core, which is typically a $\delta^{18}$O record that spans the entire length of the stack with high resolution, low noise, and no apparent hiatuses. Any problems in the signal of the target core could propagate to create errors

in core alignments and the average $\delta^{18}$O value of the stack. In the LR04 global stack, the authors checked for such errors by performing pairwise alignments to multiple target cores and comparing the stacks (Lisiecki and Raymo, 2005); however, this is a laborious process and requires subjective evaluation. Because $\delta^{18}$O variability is not globally synchronous (Skinner & Shackleton, 2005; Labeyrie et al., 2005; Waelbroeck et al., 2011; Stern & Lisiecki, 2014; Lund et al., 2015), Lisiecki and Stern (2016) created regional stacks and used a different alignment target for Atlantic versus Pacific cores.

195        The sensitivity of stacks to the choice of a single alignment target can be mitigated by aligning to a target that incorporates information from all cores in the stack. HMM-Stack (Ahn et al., 2017), which models the stack using a profile Hidden Markov model (HMM), begins with an initial alignment to a user specified target and then aligns all cores to an iteratively updated stack, which is optimized to fit all cores in the stack. Here we present a new stack construction algorithm which offers several improvements to HMM-Stack, including the opportunity to simultaneously incorporate age constraints

from all cores during the stacking process.

## 3 Methods

### 3.1 Bayesian Framework

       BIGMACS probabilistically constructs realistic age models and stacks by combining information from age proxies and stratigraphic alignment with the prior model of sedimentation rate variability from Lin et al., (2014). In Bayesian

statistics, the age information from proxy data are termed likelihoods. Specifically, likelihoods return the probability of observing the age proxies given the proposed age model and the set of model parameters. Here we refer to likelihoods as the emission model. Simply stated, the emission model returns the probabilities of residuals (or misfit) between observed data and estimated values from a particular age model. The emission model for each proxy (radiocarbon, $\delta^{18}$O, and additional age information) is discussed in section 3.3 and detailed formulations are given in the supplement (S2 and S4.1).

210        The prior model represents our *a priori* understanding of sedimentation rate variability and is termed the transition model. The transition model calculates the probability of a simulated sequence of sedimentation rates, independent of the proxy data, as described in section 3.2 and the supplement (S1 and S4.1). The transition model probabilities for a particular depth in the core are calculated as a function of both sedimentation rate change and normalized sedimentation rate (i.e., sedimentation rate expressed as a ratio the core's estimated mean sedimentation rate), given model parameters which are

derived from the same sedimentation rate data as Lin et al., (2014).

       The posterior distribution is calculated using Bayes' rule and is proportional to the product of transition and emission models. The posterior distribution of a multiproxy age model includes likelihoods returned by the radiocarbon emission model, the benthic $\delta^{18}$O emission model, and the additional age emission model. Because there is no closed form for this posterior distribution (i.e., it is not known), we employ a sampling approximation. To improve computational

efficiency, we sample the posterior using a combination of the particle smoothing (Doucet et al. 2001; Klaas et al. 2006) and Metropolis Hastings algorithms (Metropolis et al. (1953); Hastings (1970); Martino et al. (2015); section 3.4).

       In Bayesian statistics, the parameter of interest (in this case the age of sediment at a given depth) is represented by the posterior distribution, rather than a single value. Therefore, a Bayesian 95% credible interval spans 95% of the central portion of the posterior distribution. This is compared to a frequentist 95% confidence interval, which posits that there is a

95% chance that the limits are correct and encapsulate the true value. Here the 95% credible intervals and the median age model are defined by the distribution of Monte Carlo samples drawn from the posterior distribution.

The stacking algorithm is completed in two steps: an age model construction step in which a set of $\delta^{18}O$ records are aligned in parallel to a target stack (as described above), and a stack construction step in which a nonparametric regression is performed across the $\delta^{18}O$ data on the set of aligned cores. These two steps are performed iteratively until convergence. The

alignment target during age model construction is the stack from the previous iteration; for the first iteration, an initial target stack is provided by the user. The stack construction process is described in more detail in section 3.5 and S5.

**3.2 Transition Model**

For a given age, the transition model calculates the probability of the normalized sedimentation rate and the change in sedimentation rate from the previous depth (for a mathematical description, see S1 and S4.1). In its default mode, BIGMACS

uses the transmission model developed for the HMM-Match software by Lin et al. (2014); this study calculated the probabilities of normalized sedimentation rates with an empirically derived prior distribution fit to the observed sedimentation rates in 37 radiocarbon dated cores. Here we summarize the methods of Lin et al., (2014) to construct the prior; however, for more information see the original publication.

Radiocarbon ages were calibrated with the Marine09 calibration curve (Reimer et al., 2009) and sedimentation rates

were assumed to be constant between radiocarbon ages. To identify outliers and age reversals in a statistically robust manner, a Bchron age model (Haslett & Parnell, 2008) was constructed for each core. Sedimentation rates were calculated by interpolating between the modes of the Bchron ages at the depths of the radiocarbon measurements. The resulting sedimentation rates were only included in the final compilation if the following criteria was met: (1) the core was south of 40 degrees N if in the Atlantic (due to high latitude North Atlantic reservoir ages, Lisiecki & Stern 2013), (2) the core had an

average sedimentation rate of at least 8 cm/kyr, and (3) adjacent pairs of radiocarbon dates were between 0.5 kyr and 4 kyr apart. After the criteria was met, the compilation totalled 544 kyr of sediment from 37 ocean sediment cores (Figure 1, Table S1). The original study interpolated sedimentation rates every 1 kyr; however, we interpolate by 1 cm depth increments and fit a new log-normal mixture distribution (Figure 2). Interpolating sedimentation rates by depth correctly represents the frequency at which higher sedimentation rates are observed in the sediment archive, whereas interpolating by time over

represents frequency of lower sedimentation rates (which deposit less sediment per unit time).

Changes in sedimentation rates depend on both the current and previous sedimentation rate, and thus the previous two depths. However, because storing all sampled combinations of three consecutive depths is intractable for computation ($O(N^3)$, where N is the number of age model samples), normalized sedimentation rates are classified into three states: expansion, contraction, and steady. Expansion specifies a below average sedimentation rate which effectively stretches the

local portion of the record. Contraction specifies a higher sedimentation rate than the average, which requires "squeezing" the record during alignment to the target. If the local sedimentation rate is within 8% of the core's average, the state is classified as steady. In BIGMACS the probabilities of transitioning from one state to the other states are optimized via the Baum-Welch

Expectation Maximization algorithm (Rabiner, 1989; Dubrin et al., 1998). However users can also choose to keep these probabilities fixed using the sedimentation rate data from Lin et al., (2014).

BIGMACS allows a sedimentation rate change at every depth where there is proxy data ($\delta^{18}$O, $^{14}$C, or additional age information). However, in the case of low-resolution records, BIGMACS imposes a minimum age model resolution, which forces a sedimentation rate calculation every 15 cm. This depth interval was selected based on the depth spacing between the radiocarbon data used for the prior (Lin et al., 2014). Furthermore, BIGMACS normalizes sedimentation rates relative to a time-dependent average sedimentation rate calculated by the Nadaraya-Watson kernel regression (Langrene and Warin, 2019).

This accounts for longer scale changes in the depositional environment, which can be associated with transitions between glacial and interglacial oceanographic conditions.

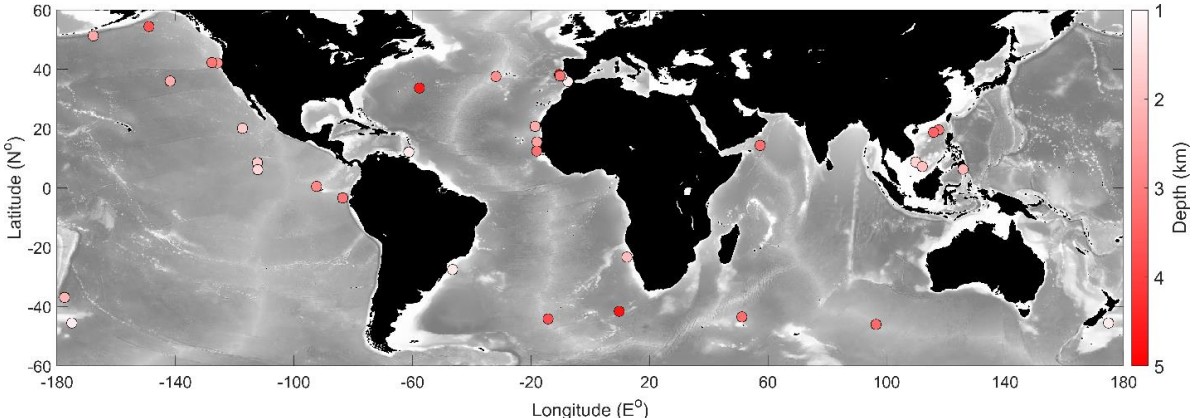

**Figure 1: Locations of cores from Lin et al., (2014) used to construct the mixed log-normal distribution.**

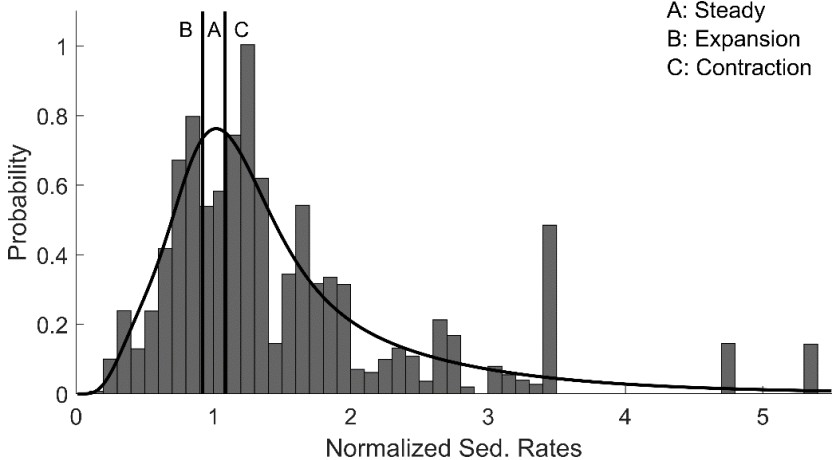


**Figure 2: The log-normal mixture fit to observed sedimentation rates from 37 cores compiled in Lin et al., (2014). Sedimentation rates are interpolated to 1 cm increments.**

## 3.3 Emission Model

BIGMACS uses different emission models for radiocarbon, $\delta^{18}O$ and additional age information (see S2 and S4.1 for more information). For radiocarbon and $\delta^{18}O$ data, the emission model is specified via generalized student's t-distributions (Christen & Peréz, 2009).

For radiocarbon data, the emission model returns the likelihood of observing age offsets from measured radiocarbon ages and depends on the radiocarbon measurement, calibration curve, and the reservoir age. The emission model also depends on two fixed parameters that control the scaling of the standard deviation. While Christen & Peréz (2009) and Blaauw & Christen (2011) set the fixed parameters of $\alpha$ and $\beta$ to three and four, we choose values of ten and eleven which produces a distribution that is more peaked and more similar to a Gaussian distribution. In other words, our student's t-distribution has smaller tails than the distribution from Christen & Perez, (2009) causing age model samples to pass closer to the mean radiocarbon age. This effectively improves agreement between the age model and the radiocarbon observations.

The $\delta^{18}O$ emission model returns the likelihood of observing different magnitudes of $\delta^{18}O$ offsets from the alignment target and depends on the target stack's time-dependent mean and variance. During alignment, Gaussian stacks are translated into a generalized student's-t distribution with the fixed parameters of $\alpha$ and $\beta$ set to three and four, respectively, based on observed $\delta^{18}O$ residuals for the ITWA and DNEA stacks (Figure S1), to address potential $\delta^{18}O$ outliers. The $\delta^{18}O$ emission model also includes core-specific scale and shift parameters which are learned across alignment iterations with the Baum-Welch Expectation Maximization algorithm (Rabiner, 1989; Durbin et al., 1998). These parameters account for vital effects among different benthic foraminifera species (e.g., Marchitto et al., 2014) and different local water mass properties at different locations (e.g., temperature and $\delta^{18}O$ of seawater). The final mean and amplitude of the stack will reflect a resolution-weighted average of the stack's component cores; thus, the average shift and scale parameters of the stacked cores will be close to zero and one (when weighted by the resolution of $\delta^{18}O$ data in each core). Optionally, the user can choose not to shift or scale individual cores during stack construction; with this setting, the variance in the stack would reflect the total $\delta^{18}O$ variance across cores.

The emission model for the additional age information (e.g., stratigraphic tie points or dated tephra layers) can either be specified as a uniform or Gaussian distribution with a mean and uncertainty specified by the user. Specifying the model as a uniform distribution will assign an equal probability for the age model to pass anywhere through the given uncertainty range. A Gaussian distribution will assign higher probabilities to age model samples that pass close to the mean of the additional age but allows for potentially larger residuals due to the tails of the distribution assigning non-zero probabilities.

## 3.4 Record Alignment

This section describes the sampling strategy employed during age model construction. Formulations for the sampling algorithm are provided in the supplement (S4.2).

Because the posterior is not given as a distribution in a closed form, age model samples are drawn using a Markov-Chain Monte Carlo (MCMC) algorithm (Peters, 2008; Martino et al., 2015). To increase computational efficiency, BIGMACS first initializes each sample using particle smoothing (Doucet et al. 2001; Klaas et al. 2006) and then refines the initialized samples with the MCMC algorithm. Particle smoothing can be understood as a continuous version of a Hidden Markov model (HMM, Durbin et al. (1998)). Whereas the HMM considers all possible hidden states because they are finite, the particle smoothing considers only a finite number of proposals because there are infinitely many possible states. In BIGMACS, the hidden states, or "particles", represent possible ages for each depth in the core. Particle smoothing consists of a forward algorithm and a backward algorithm. The forward algorithm iteratively samples and reweights particles, while the backward algorithm samples from the particles one-by-one in reverse based on their assigned weights. BIGMACS first runs particle smoothing with the state-space model defined by the transition and emission models.

BIGMACS then runs the Metropolis-Hastings algorithm (Metropolis et al. (1953); Hastings (1970); Martino et al. (2015)) to sample the proposed ages with starting points provided by the particle smoothing algorithm. The Metropolis-Hastings algorithm updates the samples block-wise, meaning that hidden states in the same sedimentation state category (expansion, contraction, and steady) are simultaneously treated in each iteration. Initialized age samples from particle smoothing allows the use of shorter chains to reach the burn-in phase.

Once the set of sampled ages are obtained, BIGMACS updates parameters of the transition and emission models via the Expectation Maximization (EM) algorithm (Dempster et al., 1977) and then iterates the process with the updated transition and emission models until convergence. If a stack is to be constructed, the final age samples are inputs to the stack construction algorithm.

## 3.5 Stack Construction Algorithm

Here we describe the Gaussian Process regression used to construct a stack construction. A formal mathematical description is presented in the supplement (section S5). During stack construction BIGMACS first aligns records to an initial $\delta^{18}O$ stack by drawing age model samples from the posterior, and then updates the stack based on the new alignments. The updated stack serves as the target for the next alignment iteration and the whole process is repeated until convergence.

A benthic $\delta^{18}O$ stack serves as a target for aligning multiple records simultaneously. Because age models are continuous, we design the stack construction algorithm to also be continuous, such that a mean and standard deviation can be defined explicitly for any age. Previous stack construction methods (Lisiecki & Stern 2016; Ahn et al., 2017) involved binning $\delta^{18}O$ data and were thus limited by the amount of data in each bin. In contrast, the continuous approach of BIGMACS allows the creation of a stack using a smaller number of records and/or with uneven data resolution over time.

BIGMACS constructs a stack using Gaussian process regression (Rasmussen and Williams, 2006), which is a continuous and nonparametric kernel-based method. In contrast to the well-known polynomial regression, a distinctive feature of Gaussian process regression is that its variance function is permitted to change along the inputs (i.e. the x-axis). BIGMACS uses the Ornstein-Uhlenbeck (OU, Rasmussen and Williams, 2006) kernel, which we find allows enough variance to resolve

millennial scale events (e.g., see sections 4.3 and 6.1.2). BIGMACS trains the OU kernel's hyperparameters, which adjust its amplitude and width, across iterations based on the data used to make the stack.

To allow the stack to reflect changes in the variance of $\delta^{18}O$ as a function of time, BIGMACS follows a heteroscedastic Gaussian process regression (Lee & Lawrence 2019) instead of a homoscedastic one. A homoscedastic Gaussian process assumes that the residuals of the data from the regression is constant but nevertheless adjusts its variance function to the proximity of data points. Thus, its variance function is narrow when data points are dense and wide where the data are less dense. A heteroscedastic Gaussian process model (used by BIGMACS) has a variance function that changes in response to the spread of the data points along inputs which allows the variance of the regression to be sensitive to the spread of responses in addition to changes in variance associated with data density from the homoscedastic Gaussian process model.

Gaussian process regressions have two major drawbacks: time complexity and outlier sensitivity. A matrix inversion, which has a time complexity equal to size of the data set cubed, is required to estimate hyperparameters for the kernel and to compute the posterior predictions. Thus, the model becomes intractable as the size of dataset increases. To address this, BIGMACS adopts the variational free energy approximation (Titsias, 2009) to make the time complexity linear to the size of dataset. Outliers are identified by the Gaussian modelling of residuals. During stack construction BIGMACS disregards outliers before performing the regression. The following two steps are iterated: 1) kernel hyperparameters are estimated after disregarding outliers, 2) outliers are classified based on the stack constructed from the estimated kernel hyperparameters.

After BIGMACS obtains a Gaussian process regression using the $\delta^{18}O$ data from every core on each sample age model, the software averages the set of regressions using moment-matching (Murphy, 2012) to produce a single Gaussian model stack in a closed form. Detailed formulations for the stack construction algorithm can be found in the supplementary note (section S5).

## 4 Results

To demonstrate the performance of BIGMACS with differing amounts and quality of data, we present two example stacks: a Deep Northeast Atlantic (DNEA) stack and an Intermediate Tropical West Atlantic (ITWA) stack. The DNEA stack is constructed using high-resolution data with relatively little noise; it consists of 2,112 $\delta^{18}O$ data points and 150 radiocarbon ages from six cores that range in depth between 2273 and 3166 m (two from the western Iberian Margin and three off the west coast of Africa). The ITWA stack is constructed from 1,066 $\delta^{18}O$ data points and 51 radiocarbon ages across four cores from the Caribbean to the northern coast of Brazil that range in depth from 1100 and 1299 m; these cores contain a relatively large number of $\delta^{18}O$ outliers. Core locations for both stacks are plotted in Figure 3. The DNEA stack spans a full glacial cycle while the ITWA stack extends to ~55 ka. We used the Deep North Atlantic (DNA) and Intermediate North Atlantic (INA) stacks from Lisiecki & Stern (2016) as initial targets for the DNEA and ITWA stacks, respectively. Default settings were used to construct both stacks. Additionally, we construct radiocarbon-only and $\delta^{18}O$-only age models for each input core to compare with the stack's multiproxy age models.

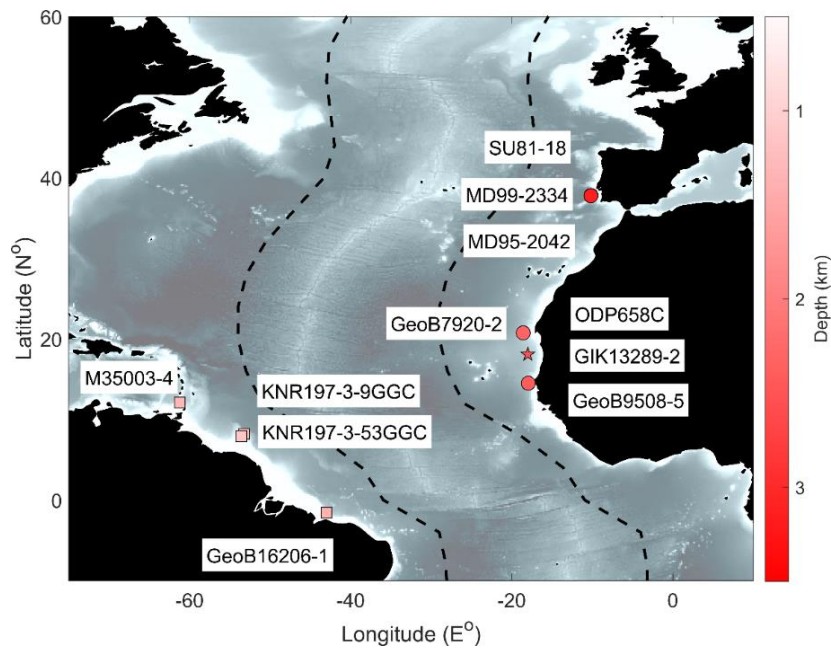

**Figure 3: Cores used to construct the DNEA stack (circles) and the ITWA stack (squares). A star indicates the core for which we use the DNEA stack as the alignment target. Dotted lines indicate east and west transects plotted in Figure 4.**

375

| Core | Lat °N | Lon °E | Depth m | ¹⁴C Citation | δ¹⁸O Citation |
|------|--------|--------|---------|------------|-------------|
| **DNEA** | | | | | |
| MD95-2042 | 37.80 | 349.83 | 3146 | Shackleton et al., 2004; Bard et al., 2017 | Shackleton et al., 2000 |
| MD99-2334 | 37.80 | 339.83 | 3166 | Skinner et al., 2003; Skinner & Shackleton., 2004; Skinner et al., (2014); Skinner et al., (2021) | Skinner & Shackleton, 2005 |
| SU81-18 | 37.77 | 349.82 | 3135 | Vogelsang et al., 2001; | Waelbroeck et al., 2001 |
| GeoB7920-2 | 20.75 | 341.42 | 2278 | Collins  et al., 2011 | Tjallingii et al., 2008 |
| ODP658C | 20.75 | 341.42 | 2273 | deMenocal et al., 2000 | Knaack & Sarnthein, 2005 |
| GeoB9508-5 | 14.5 | 342.05 | 2384 | Mulitza et al., (2008) | Mulitza et al., (2008) |
| **ITWA** | | | | | |
| M35003-4 | 12.09 | 298.76 | 1299 | Hülls & Zahn, 2010 | Hülls & Zahn, 2000 |
| KNR197-3-53GGC | 8.23 | 306.77 | 1272 | Oppo et al., 2018 | Oppo et al., 2018 |
| KNR197-3-9GGC | 7.93 | 306.42 | 1100 | Oppo et al., 2018 | Oppo et al., 2018 |
| GeoB16206-1 | -1.58 | 316.98 | 1367 | Porthilo-Ramos et al., 2017 | Voigt et al., 2017 |
| **Example** | | | | | |
| GIK13289-2 | 18.07 | 341.99 | 2485 | Sarnthein et al., 1994 | Sarnthein et al., 1994 |

**Table 1: Core locations and data citations.**

**4.1 Core Selection and Assessing Homogeneity**

When choosing alignment targets or a population of cores to construct a stack, we suggest that researchers evaluate core locations with respect to water mass reconstructions and directly compare the features of the $\delta^{18}O$ time series to evaluate whether the algorithm's assumption of homogeneous $\delta^{18}O$ variability is reasonable. Before constructing a regional stack, the user should select cores evaluated to have homogeneous $\delta^{18}O$ signals or similar water mass histories. Figure 4 shows model estimates of the fraction of Southern Component Water (SCW) in two Atlantic transects, during the present day (coloured contours, Gebbie & Huybers, 2010) and at the LGM (solid black line, Oppo et al., 2018). Here SCW refers to water that formed in the Antarctic and sub-Antarctic regions defined by Gebbie & Huybers (2010).

Core sites in the DNEA stack are just below the core of modern Northern Component Water (NCW, Figure 4) and are bathed today by 23-26% SCW and 74-77% NCW (Table S1). Glacial water mass reconstructions suggest that water mass composition at these sites was very similar during the LGM (Gebbie & Huybers, 2010; Oppo et al., 2018). A relatively constant water mass composition during the deglaciation at these sites is also suggested by neodymium isotope compilations (Howe et al., 2016; Pöppelmeier et al., 2020). Collectively, these studies support our assumption that the benthic $\delta^{18}O$ signals of these cores changed homogeneously (i.e., nearly synchronously) during Termination 1.

The cores compiled for the ITWA stack are located near the boundary between AAIW and NADW, yielding more variability in their modelled water mass percentages. SCW percentages for cores in the ITWA stack range from 31-48% for the modern and 20-28% for the LGM. During the deglaciation, AAIW experienced expansion in this region as demonstrated by a decrease in nutrients in the phosphate maximum zone (Oppo et al., 2018). Thus, the cores in the ITWA stack may have experienced moderately heterogeneous water mass changes during Termination 1. Despite moderate differences between these sites, BIGMACS is able to align these records and generate a stack that is representative of their $\delta^{18}O$ variability.

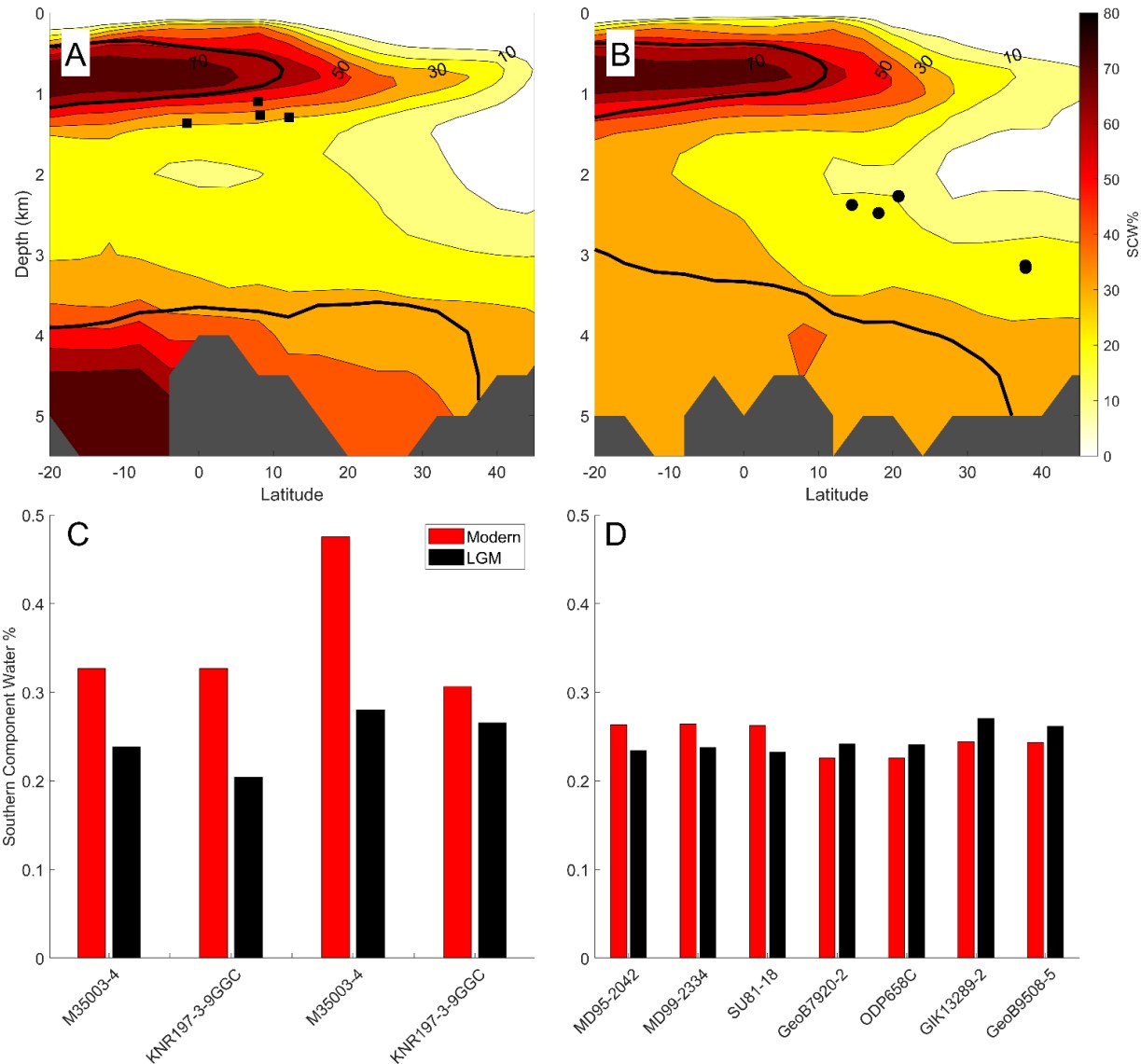

**Figure 4: (A) Western and (B) Eastern Atlantic transects of water mass composition. Transect paths are shown as dotted lines in Figure 3. Colored contours show modern Southern Component Water percentages (Gebbie & Huybers 2010) along each transect and solid black line shows the 50% contour during the LGM (Oppo et al., 2018). Solid circles represent cores in the DNEA stack, squares are cores in the ITWA stack. Histograms of modern (red) and LGM (black) southern component water percentages for cores in the (C) ITWA and (D) DNEA stacks.**

## 4.2 Age Proxies

To calibrate radiocarbon ages to calendar years, we use the Marine20 calibration curve (Heaton et al., 2020), a constant reservoir age offset (ΔR) equal to zero, and a reservoir age standard deviation of 200 years (although it should be

noted that future users can find potential reservoir age offsets using the Calib database; Reimer & Reimer, 2001). We make no corrections for the different planktonic species used to measure radiocarbon in each core (see Table 1 for data citations).

For the longest core in each stack, we provide additional age information (crosses in Figures 5A and 6A) beyond the last radiocarbon date. MD95-2042 in the DNEA stack is constrained with ages from Lisiecki & Stern (2016) identified based on an alignment of the alkenone-based SST record (Pailler & Bard, 2002) to a synthetic Greenland $\delta^{18}$O record on a speleothem age model (Barker et al., 2011; Barker & Diz., 2014). M35003-4 in the ITWA stack is constrained by an age estimate of 55.4 ka BP at 9.5 m depth based on the alignment by Hülls & Zahn, (2000) of variations in N. *dutertrei* and $CaCO_3$ to Dansgaard/Oeschger events in the GISP2 $\delta^{18}$O record (Grootes & Stuiver, 1997). This additional age information is modelled using Gaussian distributions with the standard deviations reported in Lisiecki & Stern (2016) for MD95-2042 and a standard deviation of 1 kyr for M35003-4.

## 4.3 Stack Results

Figure 7 compares the DNEA and ITWA stacks. The ITWA stack is, on average, 0.56 ‰ lighter than the DNEA stack due to the differences in deep water properties at the core sites. The ITWA core sites which span 1100-1299 m are bathed by warmer and less saline waters than the DNEA cores from 2273-3166 m. The time-dependent standard deviation in each stack (defined by the distribution of Gaussian Process regressions) reflects the variance in the aligned $\delta^{18}$O records. Between 0 and 60 ka BP, the average standard deviation is 0.13 ‰ in the DNEA stack and 0.2 ‰ in the ITWA stack. In particular, the ITWA stack has larger standard deviation during the termination, which reflects anomalously high $\delta^{18}$O values during the deglaciation in some of the ITWA cores. For example, many of the records in the ITWA stack include several anomalously high $\delta^{18}$O values during the deglaciation; Oppo et al., (2018) attributes these outliers to slope instabilities at the Demerara Rise. Because BIGMACS models a Gaussian distribution for $\delta^{18}$O residuals, the outliers produce large, symmetric confidence intervals about the mean.

The standard deviations of the two BIGMACS stacks are both smaller than the DNA and INA regional stacks from Lisiecki & Stern (2016), which average 0.24 ‰ and 0.36 ‰, respectively. This likely stems from greater benthic $\delta^{18}$O spatial variability within the larger regions defined in Lisiecki & Stern (2016) and the application of (small) record-specific shift and scale adjustments to the DNEA and ITWA cores during stacking with BIGMACS.

The Gaussian process regression also creates smoother stacks than previous binning methods. Figure S3 compares the new DNEA and ITWA stacks with the Deep North Atlantic (DNA) and Intermediate North Atlantic (INA) regional stacks from Lisiecki & Stern (2016). The Gaussian process regression creates estimates of $\delta^{18}$O for each point in time by incorporating information from neighbouring data points, which increases the stack's autocorrelation compared to the binning procedure used in Lisiecki & Stern (2016). Given the large volume of the deep ocean, we expect changes in benthic $\delta^{18}$O to respond gradually; hence smoothing may actually increase the signal-to-noise ratio of "local" stacks with less densely sampled $\delta^{18}$O measurements and relatively few cores. Although there is a risk that the Gaussian process regression may over-smooth the

data, our DNEA stack still resolves millennial scale events. For example, Figure 5(a) shows peaks at 24, 29 and 38 kyr corresponding to approximate ages of Heinrich Events H2 to H4 (Hemming, 2004), similar to the DNA stack (Figure S3).

To evaluate the multiproxy age models of the ITWA and DNEA stacks, we compare them with radiocarbon-only and $\delta^{18}O$-only age models for each core (with inclusion of the same additional ages in cores MD95-2042 and M35003-4). We find good agreement between median radiocarbon-only and multiproxy age models for each core (panels B and C in Figures 5 and 6), indicating that the $\delta^{18}O$ alignments did not cause the multiproxy age models to stray significantly from the radiocarbon age constraints. Furthermore, the multiproxy age models have 95% credible interval widths that are on average 262 years smaller than the radiocarbon age models and 1.92 kyr smaller than $\delta^{18}O$-only age models (Figure S2).

The good agreement between the radiocarbon and multiproxy median age models also supports our assertion that the input cores for each stack share homogeneous $\delta^{18}O$ signals. If the $\delta^{18}O$ records changed asynchronously, the alignments (which rely on the assumption of synchronous $\delta^{18}O$ change) would likely cause differences between the median age estimates of the radiocarbon-only and multiproxy age models. This assertion of synchronous $\delta^{18}O$ change is also supported by the relatively small shift and scale parameters learned for each core during the stacking procedure, indicating similar $\delta^{18}O$ values across all core sites (Table S1).

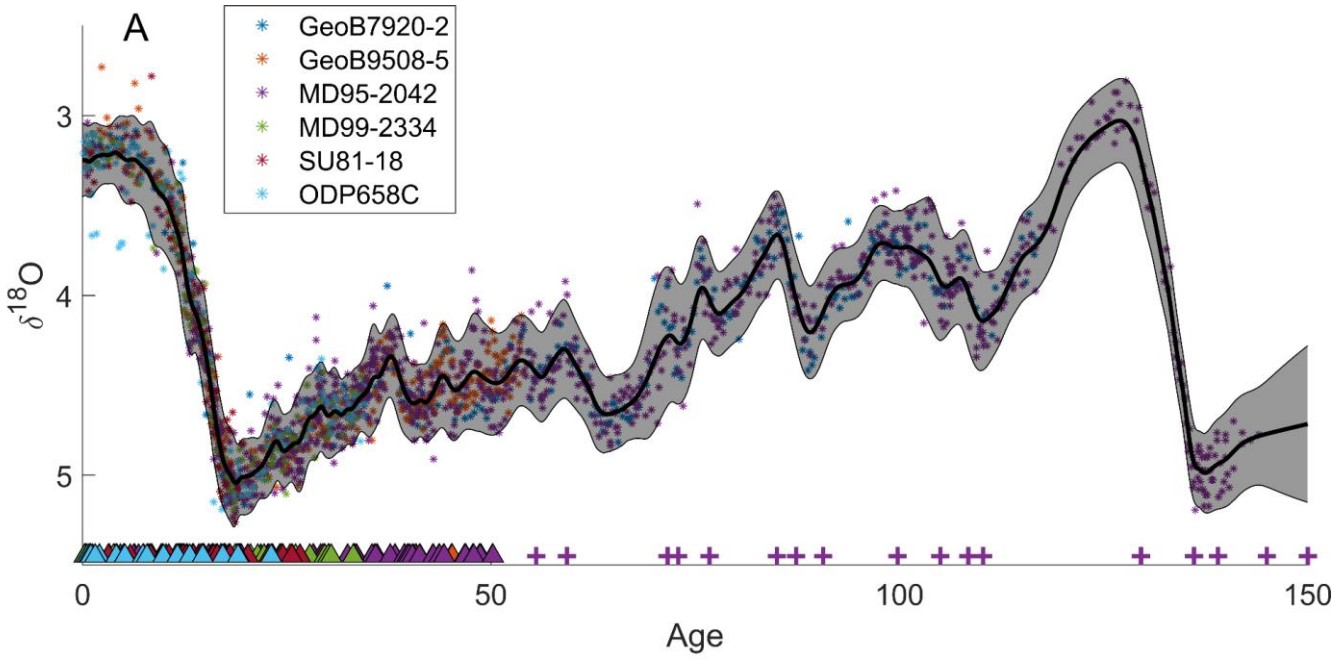

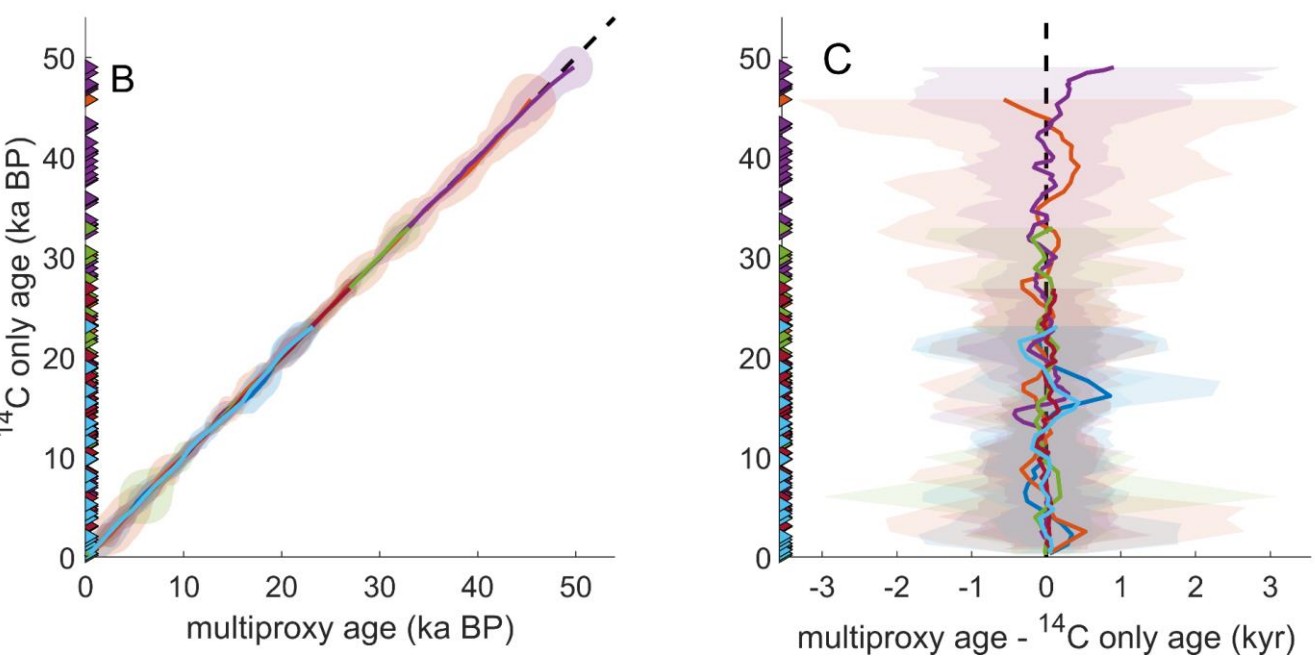

Figure 5: The Deep Northeast Atlantic (DNEA) stack. (A) The solid black line and shaded region represents the median stack value and 2-sigma upper and lower bounds. Filled circles are the shifted and scaled δ¹⁸O data points from each core on the multiproxy age models. Filled triangles mark the radiocarbon ages from the respective cores. Purple crosses are the tie points for MD95-2042 taken from Lisiecki & Stern (2016). (B) ¹⁴C-only age models vs. the multiproxy age models for each core in the DNEA stack. Each core

455

 **plots along the black dashed 1:1 line. (C) The difference between the multiproxy age models and the ¹⁴C age models for each core in the DNEA stack. Coloured shading shows the joint uncertainty distribution for ¹⁴C and multiproxy age estimates for each core.**

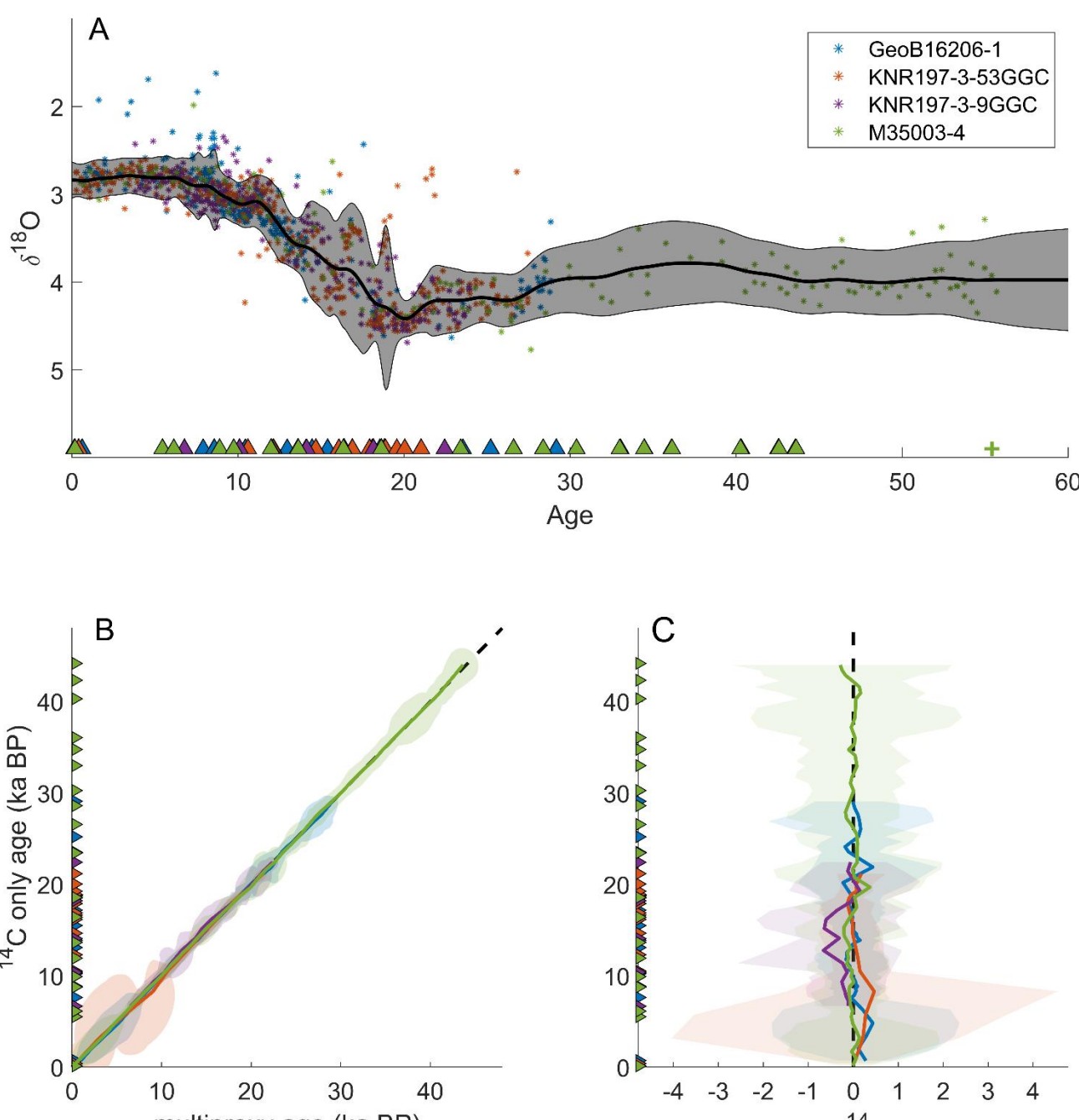

**Figure 6: The Intermediate Tropical West Atlantic (ITWA) stack. (A) The solid black line and shaded region represents the median stack value and 2-sigma upper and lower bounds. Filled circles are the shifted and scaled δ¹⁸O data points from each core on the**

**multiproxy age models. Filled triangles mark radiocarbon ages from the respective cores. The green cross is the tie point for M35003-4 from Hulz et al., (2000). (B) $^{14}$C-only age models vs. the multiproxy age models for each core in the ITWA stack. Each core plots along the black dashed 1:1 line. (C) The difference between the multiproxy age models and the $^{14}$C age models for each core. Coloured shading shows the joint uncertainty distribution for $^{14}$C and multiproxy age estimates for each core.**

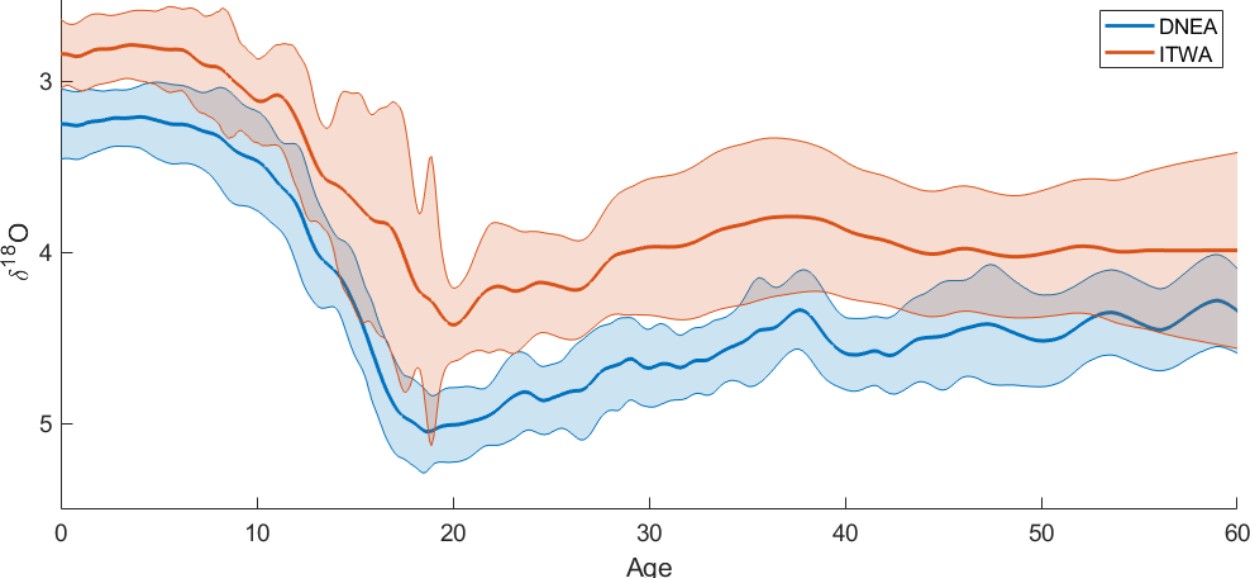

**Figure 7: Comparison of the DNEA and ITWA stacks. Median values are displayed as the thick solid line, and shading marks plus and minus two standard deviations.**

## 5 GIK13289-2 Age Model Comparison

To further evaluate the differences between single proxy and multiproxy age models, we compare three age models for GIK13289-2 constructed by BIGMACS: a radiocarbon-only age model, a $\delta^{18}$O-only age model, and a multiproxy age model constrained by both $\delta^{18}$O and radiocarbon data (Figure 8). The alignment target for the multiproxy and $\delta^{18}$O-only age models is the DNEA stack. While the radiocarbon and multiproxy age models have direct age constraints via radiocarbon ages, the $\delta^{18}$O-only age model provides only relative age constraints. Furthermore, the uncertainty for the $\delta^{18}$O-only age model reflects only the *alignment* uncertainty. The absolute age uncertainty would be a combination of the alignment uncertainty and the absolute age uncertainty from the DNEA stack.

The multiproxy and radiocarbon-only age models show similar median ages. However, the radiocarbon age model has larger confidence intervals between core depths of 1.7 and 2.2 m where there is a ~10-kyr gap between radiocarbon measurements. The multiproxy age model is constrained by five $\delta^{18}$O data points between these depths which serve to decrease age uncertainty. At a depth of 2 m, the 95% credible interval width for the multiproxy age model (5.0 kyr) is 3.8 kyr smaller than the 95% credible interval width for the radiocarbon age model (8.8 kyr).

The $\delta^{18}$O-only age model for GIK13289-2 is based only on $\delta^{18}$O alignment and has considerably larger uncertainty than the multiproxy age model, with a 95% credible interval width as much as 6.6 kyr larger. Furthermore, there is

disagreement between the median age models during the Holocene, with a maximum age difference of 2.2 kyr. The apparent

490  error in median age estimates from δ¹⁸O-only alignments likely results from near-constant δ¹⁸O values during the Holocene, which allows for more possible alignments that fit the target and a less precise age model. The 95% credible interval for the δ¹⁸O age model spans both the multiproxy and radiocarbon median ages, suggesting realistic uncertainty estimates for the alignment.

In Figure 9, the purple shading of the δ¹⁸O-based age model represents age model sample density. The non-Gaussian

495  nature of the δ¹⁸O-based age estimates is evident at the end of the age model, where the median age and darker shading are located near the upper end of the 95% credible interval. The multiproxy age model samples at this depth (which are constrained by the final radiocarbon age) agree with the dense cluster of δ¹⁸O-only age model samples. Frameworks have been developed to use the distribution of age model samples, such as those provided by BIGMACS, to estimate the probability of timing differences between climate responses recorded in multiple cores (Parnell et al., 2008; Khider et al., 2017).

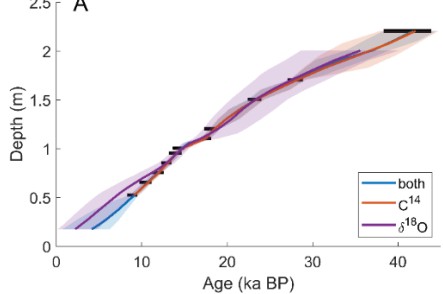

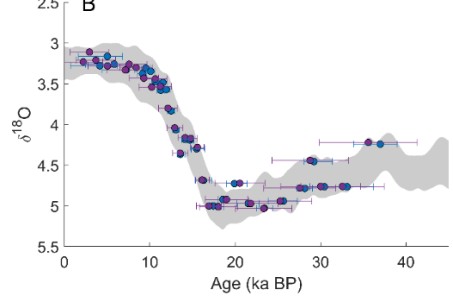

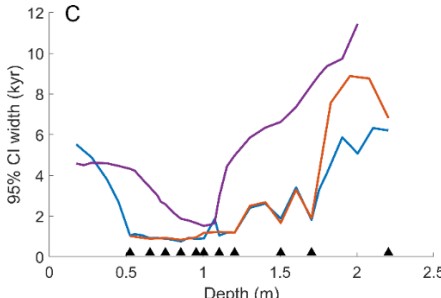

500

Figure 8: Comparison of a δ¹⁸O-only age model, radiocarbon-only age model, and multiproxy age model for GIK13289-2. (A) Age vs. depth plot, solid black lines represent calibrated radiocarbon ages. (B) The shifted and scaled δ¹⁸O for the δ¹⁸O-only age model and multiproxy age model aligned to the DNEA stack. (C) 95% credible interval widths for each age model. Black triangles indicate the depths of the radiocarbon ages. Note that the radiocarbon-only age model does not extend beyond the top ¹⁴C date of ~10 ka BP, and we do not display the ¹⁴C age model in panel (B).

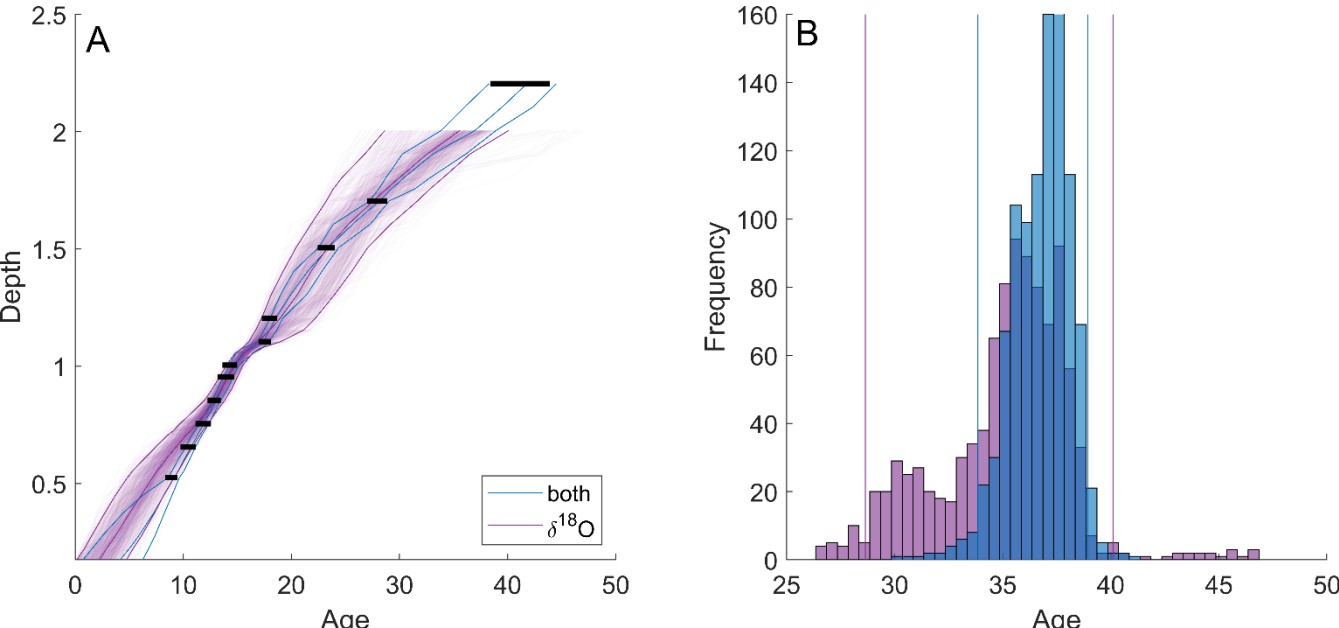

Figure 9: (A) Sample density of the δ¹⁸O-only age model for GIK13289-2. The median age model and 95% credible bands are plotted as solid purple lines. The multiproxy median age model and 95% confidence bands are also plotted (solid blue lines) along with the calibrated radiocarbon ages (horizontal black lines). (B) Histogram of δ¹⁸O-only age samples (purple) and multiproxy age model samples (blue) for the last depth in the δ¹⁸O-only age model (approximately 2 m). Vertical lines mark the 95% credible intervals at the same depth for both age models.

## 6 Discussion

### 6.1 Applications

In this section we discuss the advantages and limitations of the BIGMACS software compared to other available age modelling and stacking techniques and provide practical advice on the types of applications most suitable for BIGMACS.

#### 6.1.1 Applicability of the transition model

Most software packages which generate probabilistic age models (e.g., Bacon, Oxcal, Undatable) use models of sedimentation rate variability with tuneable parameters, which affect the amount of age uncertainty between age proxies measured at discrete depths (e.g., radiocarbon, tephra layers, tie points, etc.). During benthic δ¹⁸O alignment, sedimentation rate constraints also limit the degree to which the input record is stretched or squeezed to match the target record. In most cases, users have no specific information on which values for sedimentation rate parameters are most appropriate for the specific core analysed. Thus, parameter tuning usually increases the subjectivity and labour involved to create an age model.

Therefore, BIGMACS is designed to be used without parameter tuning. Because BIGMACS uses a prior that is constructed from a global compilation of marine sediment cores representing different environments (Lin et al., 2014; see Figure 1 and table S1), the age uncertainty returned by BIGMACS is physically realistic for most marine cores and less subjective than using tuned parameters in other software packages.

The current version of BIGMACS uses the same prior that was used in HMM-Match (Lin et al., 2014) based on a
global compilation of cores. BIGMACS can also adjust its state change probabilities based on information learned from the particular cores being aligned (see S4.3). However, BIGMACS has the flexibility to use other priors that may focus on a particular oceanographic setting or based on larger compilations of sedimentation rate variability that may be created. For example, Mulitza et al., (2021) presents a compilation of 6153 radiocarbon ages from 598 ocean sediment cores. This is potentially enough data to construct regionally specific priors if trends in the behaviours of sedimentation rates are observed
in different environments.

In addition to larger and/or more regionally focused compilations, future work includes plans to address several limitations of the method used for the Lin et al. (2014) compilation. Lin et al. (2014) used Bchron age models to identify outliers and reversals, and calculated sedimentation rates by interpolating between the mode of the Bchron age model for each calibrated $^{14}C$ date rather than the full probability distribution (see S1 for a more thorough description). Additionally, Lin et
al. (2014) used radiocarbon ages were calibrated with the Marine09 curve (Reimer et al., 2009) with $\Delta R=0$ for reservoir ages. Although we expect this to introduce relatively little bias to the sedimentation rate priors, future priors should use the updated Marine20 curve and estimates of marine reservoir ages (Heaton et al., 2020).

If users find that the default transition model does not allow enough sedimentation rate variability to fit the age proxies for a particular set of cores, it is also possible to use your own prior distribution (see the User's Manual). However, we have
not encountered such problems in testing the software, and we encourage users to exercise caution when changing this distribution.

### 6.1.2 Multiproxy age models

Multiproxy age models generated by BIGMACS provide additional advantages compared to traditional probabilistic
$^{14}C$ age models. In $^{14}C$-only age models, each core's age model is constrained only by the $^{14}C$ dates from an individual core; however, multiproxy age models can use age constraints from multiple nearby cores, which are often available for locations of particular paleoceanographic interest (e.g., cores SU81-18, MD95-2042, and MD99-2334 on the Iberian Margin). For cores sharing a similar water mass history (which is likely for neighbouring cores from similar water depths), multiproxy age models use both benthic $\delta^{18}O$ alignment and $^{14}C$ dates to generate age models for each core that are constrained by all $^{14}C$ dates in the
group of cores. This is particularly useful for cores with lower resolution $^{14}C$ dating or with ambiguous $^{14}C$ outliers. Our example of GIK13289-2 (Figure 8) demonstrates that multiproxy alignment is helpful for extending age estimates beyond the range of $^{14}C$ dates (e.g., the Holocene portion of GIK13289-2) and decreasing age uncertainty between widely spaced $^{14}C$

dates, even in cases where benthic $\delta^{18}O$ data are also relatively low resolution. In most cases, these age model benefits are enhanced when BIGMACS is used to generate a multiproxy stack (e.g., Figures 5 and 6) instead of alignment to a fixed target.

Users should be aware that the age uncertainties returned by BIGMACS for age models generated by multiproxy alignment or stacking do not include the age uncertainty of the alignment target. Thus, age uncertainties (other than those from [14]C-only mode) should interpreted as relative age uncertainties that reflect alignment uncertainty, rather than absolute age uncertainty. For multiproxy stacks constrained by densely sampled [14]C dates with small calibration uncertainty, such as the DNEA stack from 0-25 ka (Figure 5), the absolute age uncertainty of the stack will be small. However, where the absolute age

uncertainty of the alignment target or stack is larger, an assessment of a core's absolute age uncertainty should incorporate both the absolute age uncertainty of the target/stack and alignment uncertainty. For example, absolute age uncertainty for the DNEA stack beyond 45 ka can be estimated by constructing an age model for MD95-2042 using only the [14]C dates and additional age information (i.e., tie points marked as crosses in Figure 5A). Because GeoB7920-2 contains no direct age proxies beyond 45 ka, it's absolute age uncertainty could be estimated as the sum of variance in the alignment uncertainty (the age

model uncertainty resulting from alignment to the DNEA stack) and the variance of the age model constructed for MD95-2042 using only radiocarbon data and the additional tie points.

**6.1.3 Stacking**

Creating a multiproxy stack in BIGMACS offers several advantages compared to traditional stacking techniques. First, BIGMACS can create multiproxy stacks with as few as two cores. All cores in the multiproxy stack must have benthic

$\delta^{18}O$ for alignment, but the stack can include cores that lack [14]C or other age constraints. Second, whereas most previous stacks have been constructed by pairwise alignments of each core to a single target (e.g., Lisiecki and Stern, 2016), BIGMACS aligns all cores simultaneously while updating the alignment target until convergence is achieved. This process reduces the time required to create a stack as well as sensitivity to the choice of the initial alignment target. Third, the multiproxy stack's age model and alignments evolve simultaneously based on the direct age proxies in all the aligned cores, whereas most previously

constructed stacks aligned all cores before estimating the stack's age model (e.g., Huybers and Wunsch, 2004; Lisiecki & Raymo, 2005; Lisiecki & Stern, 2016). Although BIGMACS and HMM-Stack both iteratively update the alignment target using the aligned $\delta^{18}O$ signals, stacks produced by HMM-Stack implicitly inherit the age model of the original alignment target because HMM-Stack contains no procedure to input absolute age information or adjust the alignment target's age model.

Another innovation in BIGMACS is the use of the Gaussian process regression to create time-continuous estimates

of the $\delta^{18}O$ stack's mean and variance. Most previous stacks relied on either interpolation of each core's $\delta^{18}O$ measurements to an even time spacing (e.g., Huybers & Wunsch, 2004) or binning and averaging all cores' $\delta^{18}O$ measurements within a certain time window (e.g., Lisiecki and Raymo, 2005). The Gaussian process regression requires fewer cores, samples at any resolution without interpolation, smooths the stack to increase its signal-to-noise ratio, and realistically increases stack variance across $\delta^{18}O$ gaps. Learned hyperparameters of the OU kernel determine the overall smoothness of each stack and, hence, the

timescale of features that are well described by the stack. For the stacks presented here, smoothing from the Gaussian process

regression inhibits precise estimates of the amplitude and rate of change of events occurring on timescales of ~2 kyr or less. For example, the DNA stack of Lisiecki and Stern (2016), which averaged $\delta^{18}O$ values using 0.5 kyr bins, decreased by 0.47 ‰ in 1.5 kyr (from 87 to 85.5 ka) during Heinrich event 8; however, in the DNEA stack produced by BIGMACS, the $\delta^{18}O$ change is spread over an interval at least twice as long (89 to 85 ka BP, Figure S3). Additionally, although a $\delta^{18}O$ response

during Greenland interstadial 19 is recorded in both the DNA and DNEA stack at 72 ka, smoothing by the Gaussian process regression and alignment uncertainty appears to have reduced its amplitude in the BIGMACS DNEA stack.

An important caveat that applies to all $\delta^{18}O$ alignments, including BIGMACS multiproxy alignments and stacks, is that the $\delta^{18}O$ records aligned should all be homogeneous, meaning that they share the same underlying $\delta^{18}O$ signal. Because previous studies have observed temporal offsets between benthic $\delta^{18}O$ signals from core sites bathed by different water masses

(Skinner & Shackleton, 2005; Labeyrie et al., 2005; Waelbroeck et al., 2011; Stern & Lisiecki, 2014), users should only align or stack cores which share the same deep water mass history over the length of the records analysed. Whether $\delta^{18}O$ is homogeneous across core sites can, in part, be evaluated by comparing the amplitude of change and mean offset (after species-corrections) between cores. For example, BIGMACS estimates only small shift and scale differences between the cores included in the DNEA and ITWA stacks (Table S1), although large shifts are observed between the stacks. Another test is to

compare the core sites' present-day deep water mass composition and reconstructions or models of deep water mass extents at the LGM. Although glacial water mass estimates are inherently uncertain due to differences between various models and reconstructions, BIGMACS offers the flexibility to easily build different stacks to evaluate the sensitivity of results to different models of benthic $\delta^{18}O$ homogeneity.

BIGMACS may be able to align and stack proxies other than benthic $\delta^{18}O$; however, the software can currently only

align and stack one proxy at a time. For BIGMACS to accurately construct a probabilistic stack of an alternate proxy, the proxy must be homogeneous across the records in the stack with residuals that can reasonably be described with the generalized student's t-distribution that BIGMACS uses for the $\delta^{18}O$ emission model. Because the emission model is based on the variance that best describes the observations, it does not require a specific assumption about the level of noise in the measurements. However, low ratios of signal-to-noise in the proxy aligned could yield unreliable results. Preliminary analysis of planktonic

$\delta^{18}O$ alignments and stacks have yielded encouraging results, but the more heterogeneous nature of surface variability requires caution in the selection of cores which can reasonably be considered homogeneous.

The computational complexity of BIGMACS also places constraints on its applications. For the records in this study, multiproxy alignment of a single core to a target takes only 1-2 minutes while the multiproxy stacks take 1-2 hours to build on a typical desktop machine. In testing, we have successfully created $\delta^{18}O$-only and multiproxy stacks of Late Pleistocene $\delta^{18}O$

spanning the past 800 kyr, which take approximately 12 hours to run. However, we have not yet evaluated the performance of BIGMACS for records longer than 800 kyr. For a more detailed discussion of the time complexity for BIGMACS, see supplemental text S6.

# 7 Conclusion

The new software package, BIGMACS, constructs multiproxy sediment core age models and benthic $\delta^{18}O$ stacks constrained by radiocarbon ages, $\delta^{18}O$ alignment, and additional age constraints. BIGMACS requires no parameter tuning and uses an empirically derived prior model of sedimentation rate variability specific to the marine depositional environment. Radiocarbon ages are modelled using a student's t-distribution, following the methods of Christen and Peréz (2009). BIGMACS also constructs time-continuous stacks using Gaussian process regression and requires fewer cores than traditional binning methods. This facilitates building stacks for more localized regions using as few as two cores from within a homogeneous water mass as assessed by deep water mass reconstructions and/or evaluation of the estimated shift and scale parameters for the aligned cores. Example regional stacks are presented for the Deep Northeast Atlantic (DNEA) and Intermediate Tropical West Atlantic (ITWA). The stacks' median $\delta^{18}O$ values provide well-dated regional climate signals, while the stacks' standard deviations include the effects of spatial variability, multiproxy age uncertainty, measurement noise, and, in the ITWA stack, the effects of $\delta^{18}O$ outliers likely caused by sediment disturbances. Finally, a comparison of radiocarbon-only, $\delta^{18}O$-only, and multiproxy age models for one core demonstrates that the multiproxy age model yields smaller age uncertainties, particularly between radiocarbon measurements and during the Holocene $\delta^{18}O$ plateau.

## Code Availability and Software Requirements

The software package BIGMACS (developed and tested in MATLAB R2021b) and the user guide can be downloaded from https://github.com/eilion/BIGMACS. BIGMACS requires the statistics and machine learning toolbox as well as the parallel computing toolbox.

## Author Contributions

Taehee Lee (first author with Devin Rand) conceptualized the evolution of overarching research goals, performed a formal analysis, developed the methodology and the software, and performed writing – review and editing. Devin Rand (first author with Taehee Lee) curated data, validated results, visualized the data, and performed writing – original draft preparation. Lorraine Lisiecki conceptualized the project, acquired funding, served as a project administrator, supervised, and performed writing – review and editing. Geoffrey Gebbie acquired funding, provided resources, supervised, and performed writing – review and editing. Charles Lawrence conceptualized the project, performed a formal analysis, acquired funding, served as a project administrator, supervised, and performed writing – review & editing.

## Competing Interests

The authors declare that they have no conflict of interest.

## Acknowledgements

This work was supported by the National Science Foundation (NSF) under the following project numbers: OCE-1760838, OCE-1760878, and OCE-1760958.

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

Williams, C.: Gaussian Processes for Machine Learning, 66, n.d.

