# Peer review of "Bayesian age models and stacks: Combining age inferences from radiocarbon and benthic $\delta^{18}O$ stratigraphic alignment"

_EGUsphere, 2022_

## Referee Comment (RC2)

**Overview**

This is an interesting and well-presented paper which I enjoyed reading. I would like to thank the authors for their work. They provide a statistically-rigorous approach to combine information between various sediment cores when all these cores provide observations of the same (fairly smooth) underlying function. This is known as *stacking* the records.

I presume that the model builds on earlier work (called HMM-Stack, Ahn *et al.,* 2017) in its HMM aspect for each sediment core. At its heart, the method assumes that each core $j$ records the same underlying function, providing paired observations $(y_i^j, d_i^j)$ where:

$$y_i^j = f\big(\theta_j(d_i^j)\big) + \epsilon_i^j$$

Here $\theta_j(d_i^j)$ represents the age-depth relationship in each core (which can be based upon radiocarbon dating or any another technique). The methods uses MCMC to iterate between updating the age-depth models $\theta_j(\cdot)$ for each core; and the shared function $f(\cdot)$. Within this MCMC, $f(\cdot)$ is modelled as a Gaussian Process (GP), and the age-depth model in a more complex manner (presumably based upon an approach laid out in HMM-Stack). The method does some initial particle filtering but then seems to actually ditch that approach (using it only for initialisation) to use a Metropolis-Hastings As such the particle filtering appears somewhat redundant and could therefore be de-emphasized.

Overall, the paper is nicely written with sufficient technical detail to allow reproduction. The authors also give useful practical examples for $\delta^{18}O$ reconstructions from several marine cores. The method is potentially adaptable to a considerable range of scenarios and will provide a significant contribution to the community (although my expectation/experience is that for records which are not as smooth, or as shared, as $\delta^{18}O$ some bespoke modifications might be required to get the model to fit – which the authors also state).

**Statistical Comments (mainly regarding the SI):**

1) My main statistical comment is that, as a new reader, I do not sufficiently understand where the specific three state HMM age-depth model comes from. I am presuming this specific age-depth model builds on previous work. In the model, there are considered to be three states for a core. Given a particular state then the sedimentation rate follows a mixture of three log-normals restricted to being within a certain range.

   This particular sedimentation model seems extremely specific, yet its justification is not really provided in either the main paper or S1. It is not clear, to a new reader, where this model comes from: either in terms of three states (with seemingly arbitrary sedimentation rate bounds) or the mixture of three log-normals within its permitted ranges (are these fixed or also estimated). What is the benefit of such a three state model, why were the boundaries chosen, and how are the parameters for each state selected? Is it somewhat arbitrary or is there geoscientific insight as to why there are three distinct states with these values?

   I presume this model, and its explanation, comes from the earlier HMM-Stack work of Ahn et. al (2017). If so that is fine – it does not need to be re-justified here in detail. However I feel there does need to be an intuitive lay-person explanation in the Intro about how it builds on this earlier work and what is specifically new here. Currently the HMM model

appears somewhat out of nowhere. Also S1 needs much stronger referencing to that work (to clarify a reader should look there for the justification.

2) I do not quite understand how the particle filtering is used to initialise the MCMC. How do you choose which particular particles to use (after you have run the particle filtering step) as the initialisation of your later MCMC? Do you run the MCMC many times with lots of initial starting points? How have you checked actual MCMC convergence and ensured you have explored the space fully from your initialisation?

3) Outlier model – I may have misunderstood but, formally, it seems you have chosen $g()$ to depend upon $\mu$. If so, I think you probably cannot entirely ignore those observations classed as outliers in the MCMC updating. When you update the GP $\mu|O, Y, A$ I would presume that the outliers will still inform as they come from a distribution that depends upon the parameter you wish to update. Consequently, I'm not sure that formally you can ignore all the values with $O = 1$ and just fit a GP to the others.

   This is unlikely to make much difference in practice **so I am not saying that you need to change it** (but you should perhaps mention this is an approximation). Perhaps you could get around this by keep the same mean for the outliers as the stack but just altering/increasing the variance for the outlier component $g()$. If you do this then one would presumably still include the observations in updating the GP stack but the outliers would have less weight.

4) The section on length/complexity does not really tell me anything practically useful, e.g. the DNEA stack has a run time of 1.8 hours. That's partially interesting, but how many MH iterations actually is that (bearing in mind you have ditched the particle filtering by that point)? You could presumably make it arbitrarily faster/slower entirely dependent on how many iterations you run everything (optimisation, particle filters, MCMC, …). Please tell me how many MCMC iterations you performed.

**More General Applied/Presentation Comments:**

1) I think it would be worth mentioning how your work links with/alongside broader *errors-in-variables* regression analysis. Fundamentally, that is rather analogous to what you are doing here – if the primary interest is in the stack rather than the age-depth model of each core which it seems to be. In errors-in-variables analysis, one has a series of observations y where $y_i = f(\theta) + \epsilon_i$ but you do not know $\theta_i$ precisely (you only observe $T_i = \theta_i + \eta_i$). This is effectively your situation - where your sediment cores provide a specific type/structure of calendar age uncertainty $\eta_i$ and is some cases the $T_i$'s are not observed at all.

   In a geoscience setting, using Bayesian techniques similar to you, this is basically what we do to make the IntCal curves (e.g. Heaton et al. 2020) but there is also quite a lot of general statistical methodological literature (e.g. Bayesian approach of Cook and Stefanski, 1994) on the topic. Additionally, there is quite a lot of literature on *registration* in functional data analysis which could briefly be mentioned (e.g. book by J Ramsay and B Silverman).

   I also did some work with a similar (but identical) goal – aiming to sharing age information between records using tie-points and a GP – in Heaton *et al.* (2013). This was used to create calendar ages for the Pakistan and Iberian Margin (Bard et al. 2013) , and Cariaco Basin

(Hughen & Heaton 2020) data which then went into IntCal13 and IntCal20. This work was somewhat different in that we only tried to transfer dating information from one record to another and only used the tie-point ages. However it does provide a previous context where tie-points are used in a method that aims for statistical rigour rather than eye-balled tuning (with uncertainties on the contemporaneity of the ties rather like your model). Your work is however more in depth and generalisable than ours (we needed fairly simple age-depth models with multivariate covariances so owe could input then into the main IntCal curve creation)

Suggest that all this only needs a brief line or two in the Intro – just to add more detail/context about how your work fits within the wider statistical research literature.

2) My colleagues (when I tried to suggest a similar approach to them to map all features across records for other proxies) were very cautious. They felt that, for many records, the entirety of the proxy could not be mapped between cores. They rather believed that, for many proxies, it was often only the sharp/main transitions that were shared between records and they did not want to match everything.

I feel this point, that users must consider if trying to match every feature is something that will work for their proxy/data, should be made very explicitly. You do mention this in the manuscript but it is somewhat hidden and only appears towards the end (in the middle of the section on Strengths/Weaknesses on lines 520-525). I feel this caveat needs to be made considerably more prominent in the Intro/Conclusion when discussing GPs so readers will not misunderstand.

I am not a sufficient expert here, but it may be that benthic $\delta^{18}O$ is more globally homogeneous than many other proxies (and the method must be used with considerable caution for some other proxies where responses can be antithetic).

3) Your Marine sites are very spread out and will not be expected to have the same regional offset $\Delta R$ from one another. You have chosen a mean of $\Delta R = 0$ for all the cores but then quite a large uncertainty ($\sigma = 200$) on $\Delta R$ to account for uncertainty. Again this is probably fine, as you have chosen a fasirly large value (and I think everything will be somewhat led by the fitting of the many $\delta^{18}O$ measurements anyway). However, I would suggest that you might advise users to initialise a different $\Delta R$ for each core using the Reimer and Reimer (2017) database.

We do not advise people to choose $\Delta R = 0$ if they have other information available. The belief is that, at least during the Holocene, any regional $\Delta R$ will remain roughly constant over time and so will be applicable along the core (as regional upwelling/ocean depth might remain relatively constant). If you choose an independent $\Delta R$ from one observation to the next then you do not model dependence in

**Note: This is a fairly minor point that I doubt will affect your results due to the volume of $\delta^{18}O$ observations. If it is a lot of work (or the marine core sites you use do not have $\Delta R$ estimates) then I suggest you just add a caveat/explanation for the paper (rather than redo everything).**

4) Is there a reason as to why the sedimentation rates of Lin et al. (2014) are applicable elsewhere? This seems like a considerable assumption. Hence while it is potentially a strength of your method to provide automated selections of sedimentation rates it is also a considerable danger if other use it as a black box when it is not appropriate. At the very least, you must ensure that any user inputs their data on the same measurement scale (i.e. m or cm) as the analysis you did for Lin et al. (2014).

**Smaller Specific Points:**

**Main Document:**

1) Line 66 – it is not only $^{14}$C production rate changes which cause variations in past atmospheric $^{14}$C/$^{12}$C levels but also rearrangements of the carbon cycle (see e.g. Heaton *et al.* 2021). Suggest minor rewording to acknowledge this.

2) Line 370 are your stack estimates smoother because you use a GP which is fundamentally a significant smoother? Or due to other factors such as averaging over calendar ages? Also does the smoothed version lose genuine features - are the features you say you smooth/lose thought to be genuine phenomena?

3) Figure 3 and Figure 4 – in the panel As showing the final stack, can you overlay the posterior mean estimate on top of the observations (rather than underneath where currently it can't be seen)

4) Line 473-474 - *Users should be aware that the age uncertainties returned by BIGMACS for age models generated by multiproxy alignment or stacking do not include the age uncertainty of the alignment target.* I do not understand this comment about an alignment target – based upon your SI you suggest you can use your method on records where there is no a priori alignment target (i.e. when you just have a selection of cores each with their own 14C dates). Have I misunderstood?

**Suppl. Information**

1. There are repeated uses of sigma to mean many things – unclear what the values that are updated in S4 refer to. Equally what are the h's – need to be made somewhat clearer?

2. More detail is needed on the parameter choices for the age-depth model – can refer to other work if this is suitable.

3. Minor point – the likelihoods are not probabilities (the densities are continuous)

4. S5 – There is some referencing to other sections that has gone wrong: "The stack construction algorithm first iterates steps in subsections S4.2, S4.3 and S4.4 until convergence and then update the new one by the method in S4.1."

   There is no S4.4. Also, do you mean S5.1 at the end rather than S4.1?

   **General Questions (as I'm interested – not requiring further work):**

1) I tried work on a similar topic a few years ago. I found that the lack of homogeneity in the underlying functions we considered (and that it was only some features that were shared) made the method hard to implement in practice. I didn't get it to work very well (hence it remains unpublished).

   Do you think that there is something special about the $\delta^{18}O$ signals you use that mean the features are highly shared between cores? Do you expect it to work as well for more challenging/variable functions/proxies? Do you think there is a danger that you get into highly multi-modal fits in some cases which the MCMC will not fully explore – or is your age-depth prior sufficiently strong to avoid that?

2) How much of a difference do the $^{14}$C measurements really make a difference when you have to match 2000 $\delta^{18}O$ observations? Do these swamp the independent calendar age information? Might there be use in having a dependency in the proxy measurements you wish to construct (from one observation to the next)?

3) As a statistician, I think it is a bit of a shame that all of the material on the methods itself has been moved to the SI. I appreciate I am biased and that many readers will be much more interested in the results than technical details.

**References**

Ahn, S., Khider, D., Lisiecki, L. E., and Lawrence, C. E.: A probabilistic Pliocene–Pleistocene stack of benthic δ18O using a profile hidden Markov model, 2, https://doi.org/10.1093/climsys/dzx002, 2017.

Bard E., Ménot G., Rostek F., Licari L., Böning P., Edwards R.L., Cheng H., Wang Y., Heaton T.J., 2013. Radiocarbon calibration/comparison records based on marine sediments from the Pakistan and Iberian margins. Radiocarbon 55,1999-2019.

Heaton, T., Bard, E., Hughen, K., 2013. Elastic Tie-Pointing—Transferring Chronologies between Records via a Gaussian Process. *Radiocarbon, 55*(4), 1975-1997. Doi:10.2458/azu_js_rc.55.17777

Heaton T.J., Blaauw M., Blackwell P.G., Ramsey C.B., Reimer P.J., Scott E.M., 2020. The IntCal20 approach to radiocarbon calibration curve construction: a new methodology using Bayesian splines and errors-in-variables. Radiocarbon 62,821-63.

Heaton T.J., Bard E., Ramsey C.B., Butzin M., Köhler P., Muscheler R., Reimer P.J., Wacker L., 2021. Radiocarbon: A key tracer for studying Earth's dynamo, climate system, carbon cycle, and Sun. Science 374: eabd7096.

Hughen K.A., Heaton T.J., 2020. Updated Cariaco Basin $^{14}$C calibration dataset from 0–60 cal kyr BP. Radiocarbon 62,1001-43.

Reimer, R. W. and Reimer, P. J.: An Online Application for ΔR Calculation, Radiocarbon, 59, 1623–1627, https://doi.org/DOI: 10.1017/RDC.2016.117, 2017.

Cook, J. R. and Stefanski, L. A. (1994). Simulation-Extrapolation Estimation in Parametric Measurement Error Models. Journal of the American Statistical Association, 89(428):1314{1328.

---

## Author Comment (AC1)

We thank Reviewer 1 for the constructive and thorough comments, especially regarding the implementation of the open-source software and the detailed software review. A point-by-point response to each comment is detailed below (blue).

This paper presents an interesting evolution of the the previous d18O alignment and stacking functions. The main advancements are the the use of an empirically derived sedimentation rate prior and incorporation additional age information including radiocarbon dates and tephras, tie points, etc. While I think this will be a fine contribution to the field, I think the manuscript needs to be further clarified. The bulk of the background focuses on d18O alignment. The radiocarbon descriptions can be somewhat abbreviated and overly simplistic.

The main issue I have is that the information from which the prior was obtained is completely absent. The reader knows nothing about the sediment cores, their locations, age ranges, and depositional environments. An empirically derived prior that replaces tunable parameters is only an advancement if it is appropriate to the readers sediment core. Statements around Line 220 seem to indicate that the prior is a poor match for the data, and the prior, not the radiocarbon dates are the greatest influence on the age model in the radiocarbon-dated interval. Hence, I recommend the discussion of the prior be greatly enhanced.

Thank you for bringing this to our attention. The prior presented here is derived from the same compilation of cores as Lin et al., (2014), i.e., HMM-Match. We recognize the importance of transparency, and in the revisions, we will include a thorough description of these cores, criteria for inclusion, and the methods of construction. In addition, we will add a section in the discussion that describes the potential strengths and weaknesses, the appropriateness of this data to be used as a prior for any ocean sediment core, and the option of users to substitute alternate priors. Also, our statement on line 220 was unclear and has been misinterpreted; see below.

I suggest the authors take a close look at spellings of acronyms and names. I noticed several different (and incorrect) spellings of Obrochta, as well as some acronyms that were transposed. I've noted the former in the below line-by-line comments.

Thank you, we will fix this.

Finally, the manuscript needs to state the system requirements for running the software. I note that it uses a parallel for loop, which is not included in the standard matlab distribution. So users without the Parallel Computing Toolbox cannot use this. Also of course the sampling of PDFs will require the stats toolbox. In my experience, most people have the stats toolbox but fewer can run a parfor loop.

While the system requirements are currently stated in the User's Manual, we will add this to the manuscript as well.

\*\*\*

Line by line comments

Lines 9 - 11: "...designed to use either age proxies (e.g., radiocarbon or tephra layers) or stratigraphic alignment (e.g., of benthic δ18O) and cannot combine age inferences from both techniques."

This is a bit misleading because other Bayesian models can indeed use oxygen isotope information -- just not in the way that is done in the present paper. A manually identified oxygen isotope tie point to a reference series can be used together with radiocarbon or tephra, so I suggest this first sentence be reworded.

We will revise this sentence to clarify that this refers to automated alignment.

Lines 28 - 30: "However, this method is restricted to the last 50 ka BP, suffers from variable surface reservoir ages ..., and is often low resolution causing the age model to be highly dependent on assumptions regarding sediment accumulation rate variability."

This is a somewhat outdated viewpoint. An increasing number of labs have installed AMSs, advances in automation have reduced preparation time and cost, and the ability to reliably measure radiocarbon on trace amounts of samples is making it increasingly possible to perform high resolution radiocarbon dating. Also, technically, the current marine calibration curve now extends to 55 ka, not 50 (though I agree that radiocarbon dating at those extreme ages is problematic, and resolution does not match oxygen isotope data). One study that comes to mind is:

Ishiwa, T. et al. Temporal variation in radiocarbon pathways caused by sea-level and tidal changes in the Bonaparte Gulf, northwestern Australia. Quaternary Science Reviews 266, 107079 (2021).

We will revise the sentence to reflect the 55 ka BP end age of Marine20. In addition, we will revise the text to simply state that radiocarbon is generally lower resolution than benthic d18O.

Lines 36 - 37: "Software packages exist to produce probabilistic age models using radiocarbon ages (Blaauw & Christen 2011; Lougheed & Obrachta, 2019), but none of these probabilistically combine age inferences from both dating techniques."

This description should be improved to clarify that the authors are referring to software packages that automatically find the optimal alignment to a reference series. The current descriptions reads as if none of the radiocarbon-centric models can use oxygen isotope tie points.

The sentence will be changed to clarify that it refers to continuous probabilistic alignment rather than tie point identification.

Line 46: Beyond 55 ka

This will be changed from 50 to 55 ka BP.

Lines 48 - 49: "Sedimentation rates are realistically constrained with an empirically derived prior model rather than subjective parameter settings."

I wonder how appropriate this prior is for the possible range of sediment cores that users will inevitably throw at your model? Is it possible to specify your own priors?

It is possible to specify your own prior, which we will specify in the revised text. However, alternate priors should be based on observational data or physically realistic models of the sedimentation process. While some users won't have the interest or specialized knowledge to do this, researchers interested in these processes could publish new prior models that can be used in the software, and we have plans to develop improved priors based on additional data in the future. Another worthwhile feature to implement is the ability to have a prior that depends on the sedimentation environment. Such a prior is conceptually possible with BIGMACS. For now, the manuscript will also be revised to provide more information on the prior we use here from Lin et al., (2014). We will provide core locations and resolutions, and more thoroughly describe our method of construction.

Lines 65 - 66: "Radiocarbon ages must be calibrated from 14C years to calendar years with a calibration curve that accounts for changes in past atmospheric 14C production rates (Reimer et al., 2020; Heaton et al., 2020)"

This is a very simplified statement. Changes in the carbon cycle is also taken into account, and quite a lot of work has gone into better understanding changes in marine reservoir age for Marine20.

This sentence will be revised to reflect the processes described in Heaton et al., 2021 that were considered during the construction of the Marine20 curve.

Lines 67 - 68: "The uncertainty of the calibrated age is a combination of the calibration curve uncertainty, the radiocarbon measurement uncertainty, and the marine reservoir age uncertainty."

To this list should be added 1) local reservoir age offset from the global mean, Delta R, which also has its own uncertainty and is 2) temporally variable (e.g., older water at downstream upwelling sites following AMOC slowdown, etc.)

This will be added.

Line 76: "LGS" should be "LSG". Also perhaps Heaton 2020 should also be cited here because Marine20 includes the BICYCLE LGS-OGCM. (see above comment starting Line 65 -- Both IntCal20 and particularly Marine20 are much more sophisticated (complicated?) than just correcting for changes in production rate.

Yes Heaton et al., 2020 will be cited here and LGS will be corrected to LSG.

Lines 106 and 108: "trial and error"

As with the other models described, Undatable also comes with suggestions regarding parameter selection. Both of these sentences would probably be better without "trial and error". If it took hours to

converge, then that would be "trial and error", but since it takes seconds, It's more like adjusting music volume to one's desired level through instantly received feedback, which is not "trial and error". I'd suggest rewriting as:

"Its quick runtime encourages parameter tuning, based on the authors' recommendations"

and

"These parameters have large effects on the resulting age model requiring the user to decide on the most appropriate values rather than using a prior model of sedimentation rate variability."

And I also suggest that the tunable parameters in the other models be similarly discussed. As it is, this description reads as is undatable is the only one with parameters that can be tuned. This somewhat undersells what the authors are presenting here: a model without tunable parameters.

The tunable parameters in the other models (e.g., the beta and gamma distributions for Bacon and k parameter for Oxcal) are mentioned but perhaps not emphasized as tunable. We will improve the clarity here. We will also make clear that the authors of Undatable offer guidelines when choosing parameter values, and we will remove "trial and error" from the description.

Line 112: "which often correlates with salinity."

yes it might loosely correspond to salinity but it's really surface evaporation - precipitation prior to deepwater formation (since I assume the author's mainly considering benthic oxygen isotopes.

Yes our main consideration is benthic d18O which does often correlate with salinity.

Line 115: "The most conservative technique for aligning records to a target is to assume that large, easily identifiable features in the signals, such as glacial terminations, occurred simultaneously, create tie points between these features, and linearly interpolate between the tie points"

There absolutely is a lag between "upstream" sites in the North Atlantic and "downstream" sites since it can take on the order of 1000 years or more for the signal to propagate with the flow of deepwater.

Yes, we agree that lags exist between benthic d18O records in many cores, and we are not trying to encourage readers to always assume that benthic d18O is synchronous. In fact, in section 4.1, we specifically advise users to align only nearby cores from within the same water mass to reduce the risk of aligning diachronous signals. Here we are trying to give background and context to benthic d18O stratigraphic alignment.

Line 135: I suggest this be better presented with the information starting Line 115.

We state this at the end of the section because it applies to each alignment method. Both tie-point identification and probabilistic alignment suffer from diachronous benthic d18O signals.

Line 165: "termed the likelihoods" remove "the"

We will remove the "the".

Line 171 - 175 The sedimentation model is called a prior distribution which is in turn called a transition model. Perhaps this can be made more clear.

We will reword to state "We refer to the prior for sedimentation rates as the transition model"

Line 180: "confidence" should be "credible"?

Yes. We will change "confidence" to "credible" throughout the manuscript with the explanation upon first mention that "credible intervals" are Bayesian confidence intervals.

Line 190: When are the locations of these 37 cores going to be disclosed?

We will add a thorough description of these cores in the revised draft.

Line 191: "However, where the previous study interpolated sedimentation rates every 1 kyr, we interpolate by 1 cm"

What is the range of sedimentation rates in the 37 cores? Is 1 cm sampling typically equivalent to a 1 ky sampling, or is the interpolation interval vastly different than that used by Lin et al?

We will more thoroughly describe these cores and the calculated sedimentation rates in the revisions.

Line 201 - 202: "Expansion specifies a below average sedimentation rate and refers to a stretching of the local portion of the record."

This is a bit confusing as stated and doesn't become clear until the next sentence where the authors stake that "contraction ... requires squeezing" Maybe rewrite as:

"Expansion refers to a below average sedimentation necessitation stretching the local portion of the record"

Ok we will make this clearer.

Lines 203 - 204: "If the local sedimentation rate is within 8% of the core's average, the state is classified as steady."

How was 8% selected? Please further clarify as is done on lines 209 - 210 regarding the 15 cm interval.

We will more thoroughly describe the transition model and the choices distinguishing expansion, contraction, and steady in the revisions.

Line 220: "improves agreement between the core age models and the radiocarbon observations"

I don't understand this sentence. The age model should be based on the radiocarbon observation in the radiocarbon-dated intervals. Does this indicate that the prior is often vastly different from the data, and

without changing the alpha and beta parameters relative to the previous Bayesian models, the age model obtained by BIGMACS is inconsistent with the radiocarbon dates?

We will rephrase to make our meaning clearer. The likelihood for radiocarbon ages is a student's-t distribution which scales the standard deviation of the radiocarbon measurement. The degree of scaling depends on the values of alpha and beta. Using alpha and beta values of 10 and 11 applies a smaller scaling and a more peaked student's-t distribution that is more similar to the shape of a normal distribution. Thus, age model samples have a higher probability to pass closer to the mean of the radiocarbon measurement compared to using alpha and beta values of 3 and 4.

Lines 234 - 237: "Specifying the model as a uniform distribution will force the age model to pass through the given uncertainty range and should be used when the user is confident about the age information. Specifying a Gaussian distribution will allow the age model to pass farther from the additional age constraint."

This seems backwards to me. If I specify a tephra age as a gaussian distribution with some mean and standard deviation, the highest probability is at the mean, so the model should pass closest to the mean. But if I specify a uniform distribution, the model has an equal probability of passing anywhere. So wouldn't the user want to specify a gaussian when there is good confidence in the age constraint? Perhaps I'm not following what the authors mean to say. Is it that when there is confidence in the *other* age data, with less confidence in the specified tephra/tie point, that the authors are suggesting to use a uniform distribution? I think this statements needs to be clarified.

This is a good point. A comparison between the closeness of an age model with additional age modeled as a Gaussian vs. a Uniform is difficult to generalize for every scenario. We will revise the this statement to reflect this.

Lines 301 - 302: "these cores contain a relatively large number of δ18O outliers (Figure 1)."

Not an appropriate text location to reference fig 1. Please add lat and lon to fig 1.

We will move where we place the call out to Figure 1 and will add lat & long to the figure

Table 1: confirm the longitudes

The longitudes are in degrees East and have been rounded. We will specify °E and provide two decimal places.

Figure 2: a color bar for the panels A and B would be helpful.

Ok we will add a color bar to this figure.

Line 347: "... crosses in Figures 4A and 5A ..."

Figure 3 has yet to be mentioned. Confirm figure numberings. I think this should be Fig 3A and 4A. Generally Figures are numbered in the order they are mentioned in the text.

Thank you for catching this mistake. We will correct the figure numbers.

Line 356: "Figure 6 compares the DNEA and ITWA stacks"

Change to Fig 5.

We will fix this, thank you for catching it.

Line 370: "The Gaussian process regression also creates smoother stacks than previous binning methods"

It would be very useful to the reader to see a comparison of the previous stacking methods. It would also be very helpful to add a figure showing each sediment core's d18O record plotted in a separate panel above the BIGMACS stack. This will let the reader better visualize the the smoothing due to the increased autocorrelation. This would also support the assertion on Line 385 of homogenous signals.

Comparison between the DNEA and ITWA stacks constructed with BIGMACS and the DNA and INA stack constructed with Match from Lisiecki & Stern (2016) is presented in Figure S4. We will add a call out to that figure here. In these plots the apparent smoothing is clear.

While core-specific benthic d18O records are plotted in Figures 3a and 4a, we will add supplemental figures for each core that more clearly plot individual benthic d18O records.

Line 449: "6.1.1 Radiocarbon and multiproxy age models"

Missing from the discussion of applicability is, of course, if the goal is to compare phasing between d18O records, then the multiproxy age model cannot be used and only 14C, tephras, etc. can be used.

This is work we are currently investigating. We have developed a method to calculate lags between benthic d18O records by comparing 14C-only and d18O-only age models. The focus of this section is the advantages of BIGMACS over other age modeling software packages and we will add a description of this application.

Lines 456 - 458: "Because BIGMACS applies a prior model based on observed sedimentation rate variability (Lin et al., 2014), the age uncertainty between 14C observations returned by BIGMACS is physically realistic and less subjective than using tuned parameters in other software packages."

At this point, we still know nothing about the cores from which this prior was obtained. Where are they located? What are their water depths? What are their age ranges? Do they span glacial/interglacial terminations? While this methods does not require parameter selection, it is assuming that the prior is reasonable for the *user's* sediment cores. This is an extremely important point, and I think the authors should spend some time to demonstrate to the reader that the prior is actually appropriate. In short, I'd like to have it explained to me very clearly why the prior assumed here is both appropriate and better than selecting parameters. The statement I mentioned earlier on Line 220 gives the impression that the prior is overly informative and inconsistent with the data.

We will provide a thorough description of the cores included in the prior, the radiocarbon ages, the criteria for inclusion, and the methods to calculate the sedimentation rates. In addition, we will provide a new section in the discussion that addresses the application of this prior on a global scale. In addition, we will make clear that improvement of the prior is currently being investigated. Future versions of the software package will likely have users make a choice between multiple priors based on the sedimentation environment of their core.

Line 470: "widely space"

"spaced"

Thank you we will make this correction.

Lines 478 - 479: "an assessment of a core's absolute age uncertainty should incorporate both the absolute age uncertainty of the target/stack and alignment uncertainty."

I would suggest adding an optional age error column for the stacking target, then fold that error into the alignment uncertainty. You could output both age uncertainty obtained from that of the alignment target, in addition to the alignment uncertainty already returned. The could be added to get a total uncertainty.

This is a potential future revision of the software package, and beyond the scope of the current manuscript. BIGMACS improves upon other age modeling software packages by combining direct dating techniques with probabilistic benthic d18O alignment to construct age models and stacks. During a d18O-only alignment, the uncertainty in the resulting age model only reflects the alignment uncertainty and does not include uncertainty from the target's age model. Combining these uncertainties is an obvious next step. The process of calculating absolute age uncertainty differs for stack construction and benthic d18O alignment and would increase run times. For example, absolute age uncertainty could be calculated for alignments by generating sample alignment targets consistent with the stack/target and re-running the alignment process for each sample target. The computational cost of this calculation is such that all/most currently available automated d18O alignment software (e.g., HMM-Match) does not include this feature. We agree this would be immensely useful and hope to add it to a future version of BIGMACS.

Line 544: "Example multiproxy regional stacks"

The age models are "multiproxy" but the stacks are not.

Thank you we will clarify this.

Line 547: "standard deviations include the effects of spatial variability, age uncertainty ..."

I really think that there should be an easier way for users to include the age uncertainty in the alignment target.

This is definitely something we can address in future iterations of BIGMACS.

Author contributions: It appears that the first two listed authors contributed equally. As such they should be listed as "contributed equally" somewhere around where the corresponding authors are noted. If the other authors only contributed funding for this study, then technically they should not be authors and should be acknowledged.

Yes, the first two authors contributed and equally. We will add this to the corresponding authors list. All other authors directly contributed greatly to this work to and should be listed as co-authors. There contributions are specified accordingly under 'Author Contributions' at the end of the manuscript.
* * *
I didn't do a full code review but I do have some suggestions as the authors suggest that BIGMACS is resource intensive and slow. There are several things that I see that could be optimized. While I feel that the time and memory savings on the things I am point out will be minimal, it makes me wonder if there are similar inefficiencies in the most critical parts of code.

getInitialTarget.m

Line 66, load calibration curve

Why not just load only the curves that are needed? There are much more efficient ways to read in the data. The fastest is just remove the headerline of each calcurve and use simply load(path). Small things like this, if they occur throughout the code base, can add up to a savings in runtime. Also note that "path" is a command to Get/set search the path. I'd suggest changing the variable name to "Path" or "pth".

The reason why we did not remove the header line is because we hope to make the software more user-friendly: without it, a user might be confused in understanding what each column means. However, your note regarding 'path' is reasonable: we will reflect it to the next version of the software.

Why do all this:

*tic*

*path = 'Defaults/Calibration_Curves/IntCal20.txt';*

*fileID = fopen(path);*

*CAL = textscan(fileID,'%s %s %s %s %s');*

*fclose(fileID);*

*cal_curve{1} = zeros(length(CAL{1})-1,5);*

*for k = 1:5*

*cal_curve{1}(:,k) = str2double(CAL{k}(2:end));*

*end*

*toc*

Elapsed time is 0.390675 seconds.

when you can do simply this, which is simpler and an order of magnitude faster. Are there similar chunks of inefficient code that are resulting in slow runtime?

*tic*

*Path = 'Defaults/Calibration_Curves/IntCal20.txt';*

*fileID = fopen(Path);*

*CAL = textscan(fileID,'%d %d %d %d %d','headerlines',1);*

*toc*

Elapsed time is 0.018334 seconds.

getData.m

If you can figure out what the final size of e.g., "d18O_depth" will be, you can preallocate a matrix for better memory management and faster runtime.

Though loading files is not the bottleneck in cases that we have computed, your comment regarding an efficient coding make sense. We will reflect them to the next version of the software. However, the main limit for runtime efficiency is the use of Matlab. We do not currently have the resources to convert the MATLAB codes into Python or C++, but we hope to do this in the future.

initializeAlignment.m getAlignment.m

This function uses a parfor loop, which requires the parallel computing toolbox that not everyone will have. It will also take time to start up the parallel pool if not already running. Could check for the existence of the toolbox and if it's not installed, use for instead. If there is not a significant improvement in speed, giving the time to start up the pool, it might be better to use just a for loop.

It is true that we have presumed that users may have the parallel computing toolbox and we will clearly state the program requirements in the manuscript.

---

## Author Comment (AC3)

We thank Luke Skinner for his summaries of the reviews and insights into radiocarbon and benthic d18O age model construction. It is our belief that we can sufficiently address each concern in a revision of the manuscript. Below is a point-by-point response to the editor's comments.

Now that a sufficient number of reviews have been submitted, I would like to invite you provide a point-by-point response to each of the review comments. I would like to underline the importance of addressing a key issue that has been raised in the reviews, namely: the 'empirically constrained' sedimentation rate prior that is applied in the matching algorithm. One issue is that the validity and applicability of this prior, across a range of sedimentary contexts, does not appear to have been fully assessed in a transparent manner in the manuscript – and indeed seems doubtful.

Below I add a few remarks of my own, in case these are helpful for considering what revisions you would undertake, and I invite your response to these too. I find your study of particular interest, and I hope that my comments will be seen as useful.

Title/general ethos:

In general, I think it might be useful to more clearly delineate the distinction between alignment and 'dating' at the core of the manuscript (even though the difference between relative and 'absolute/numerical' ages is indeed noted in the paper a few times). Creating a benthic d18O stack is one thing, aligning to a benthic d18O stack is another, and dating a sediment core is yet another.

As you say, the manuscript addresses the distinction between relative and absolute ages a few times already, but we will attempt to further highlight this in our revision. Your delineation of three separate processes: alignment, stacking, and dating has been true in the past; however, our new approach integrates these three steps together, such that this is no longer the case. Dating from absolute age proxies is now incorporated simultaneously with the determination of relative ages from alignment. This is why we feel it is appropriate to call these "multiproxy age models;" it is not simply the assignment of a radiocarbon age model after alignment or stacking. We will better describe the nuance and novelty of our approach in the revised draft.

The only way that benthic alignment provides age constraints is if one proposes to have prior knowledge of how local/regional deep-water T and d18Osw relate to insolation, e.g. based on a hypothesis for how insolation paces ice volume, and how changes in ice volume are linked to deep water T changes and/or influence deep ocean d18Osw at a given location in the ocean. The latter sequence of hypotheses can give age constraints that are of ~millennial accuracy at best. In such a context, radiocarbon dates (even with ~centennial uncertainties in reservoir age offsets) obviously can provide a refinement.

BIGMACS is not attempting to use the benthic d18O as an indicator of absolute time (e.g., via orbital tuning) as you describe here. d18O is used only to constrain relative ages between cores (or a target stack). We will add more clarification of this point. Our motivation in developing this software is the same point you make above, there is added value in simultaneously considering benthic d18O and 14C during the alignment process.

The inverse is unlikely to be true: age constraints on benthic d18O are unlikely to be precise enough, even to constrain changes in radiocarbon reservoir age offsets of order 100-1000 years.

We do not intend to make such a claim in the manuscript and would welcome information regarding locations of the text that lack clarity regarding this point.

Alternatively, if the core notion of the manuscript and algorithm is the simple transferral of a radiocarbon chronology (or the pooling of radiocarbon dates) between sediment cores via a stratigraphic alignment of benthic d18O, then again it is not quite a case of 'combining age constraints from radiocarbon and benthic d18O'. Rather, it is one of radiocarbon dating of a stratigraphic alignment/stack.

The main concern here appears to be a nuance in how the age model technique is described. Your description of a "transferral of a radiocarbon chronology (or the pooling of radiocarbon dates) between sediment cores via a stratigraphic alignment of benthic d18O" is a reasonable summary of the software during stack construction. However, BIGMACS can also include other types of age constraints, offers an alignment-only mode which does not require any 14C dates, and also constrains age models based on a prior for sedimentation rate variability. While the concept is simple, the probabilistic model leads to rigorous statistical results that simultaneously combines many pieces of information.

What the software does is more complex than "radiocarbon dating of an alignment/stack" because the alignment/stacking is performed simultaneously with consideration of the 14C age information. Because the software integrates relative age information from benthic d18O simultaneously with 14C age constraints, we feel that "multiproxy" is an appropriate description of the age modeling method. In fact, we show in Figure S3 that considering both d18O alignment and 14C dates (ie, multiple proxies) in age model construction produces smaller confidence intervals than either alone (in large part due to the pooling of 14C dates from multiple cores, which requires alignments of the cores).

As such, my own feeling is that the manuscript might more accurately be framed in terms of 'refining orbitally-tuned benthic d18O age models using radiocarbon constraints', e.g. in the title and through the text.

There seems to be some misunderstanding here. No orbitally tuned ages are incorporated in BIGMACS unless a user chooses to align to an orbitally tuned target. None of our examples use any age information derived from orbital tuning. If radiocarbon dates are used in stack construction, the final age model produced will derive almost entirely from 14C over the time period for which 14C dates are available (because the stack is iteratively updated to agree with absolute age constraints).

In a similar vein, it seems to me that describing the age models as 'multi-proxy' is a little misleading: my own expectation was initially of something like that described in line 522. I would again suggest that the process tackled in the present study be described as something like 'radiocarbon-refined single proxy alignment'.

You are likely not the only reader who may initially misinterpret our use of the word multiproxy, and we concede that it is important to clearly explain our use of the term multiproxy in the abstract and early in the introduction (ie, multiple types of age information). Although currently BIGMACS can only align benthic d18O data, we hope to develop a future version will be equipped to handle alignments of multiple climate/sediment proxies.

We refer to the age models as multiproxy because information from both radiocarbon ages and benthic d18O-alignment are simultaneously integrated during age model construction. If the user has an input core with radiocarbon data, benthic d18O data, and an independently dated alignment target that they believe has a synchronous benthic d18O signal, then the combination of both proxies determine the age uncertainty of the age model. Figure S3 demonstrates that the multiproxy age models have smaller uncertainties than their single proxy counterparts for every core in this study.

BIGMACS can also incorporate "multiproxy" (relative and absolute) age information on timescales beyond the limit of 14C dating with the use of additional sources of absolute age information (called "additional ages" in the software). For example, tephra layers or tie points to speleothems can provide additional absolute age information to improve an age model, compared to a "single proxy" alignment based solely on benthic d18O alignment.

Although other age modeling software can combine absolute age information from 14C with non-14C sources, none of the other dating software combines these absolute age estimates with automated signal alignment (ie, probabilistically derived relative ages). For example, Undatable only incorporates relative age information if the user specifies an absolute age estimate for a discrete tie point. This is dramatically different from the continuous probabilistic relative (aligned) age estimates that BIGMACS integrates with absolute age constraints.

Perhaps the source of concern with the term "multiproxy" is that benthic d18O and 14C are proxies for different things. 14C is a proxy for absolute age whereas benthic d18O is a proxy for climate/seawater properties and, thus, relative age.  We are happy to add more explanation to clarify our intended meaning of "multiproxy," but we assert that our use of the term is both technically correct and appropriately conveys the integration of multiple sources of (different) age information. However, if a decision about publication of this manuscript ultimately rests on our inclusion of the word multiproxy in its description, we are willing to discuss alternatives.

Line 25:

This line is not quite correct: the accuracy with which ocean sediment cores can reconstruct the *timing* of past climate events, depends on.. the.. age model.  The accuracy of proxies is a separate (thorny) matter.

 Ok we will revise this sentence to state "The accuracy with which ocean sediment core data can reconstruct the timing of past climate events…"

Line 48:

"Sedimentation rates are realistically constrained…."

As pointed out by the first Reviewer, it seems we must take this on faith, whereas there is burden of demonstration here.

 In our revision we will include an in-depth description of the data and construction methods of the transition model. In addition, we will add a section in the discussion that summarizes the strengths and weaknesses of this prior, as well as areas of future improvements.

Line 65:

In general, there is a need to be precise when describing radiocarbon procedures.  Radiocarbon dates need to be calibrated to account for past changes in the initial radiocarbon concentration of the fossil entity's 'parent reservoir' (atmosphere, surface ocean, etc.), which may change due to 14C-production changes and/or other carbon cycle processes.  This crops up again on Line 72: planktonic foraminiferal radiocarbon dates must be corrected for 'reservoir age offsets' (relative to the atmosphere) only if using a record of past atmospheric radiocarbon concentration/activity for the calibration. In principle, a 'marine calibration curve' might be used instead, with different potential corrections needed as a result.

 Thank you for catching this. We will revise these sentences to reflect the processes described in Heaton et al., (2020).

Line 79:

"…requires simulating the core's sedimentation rate."

I think this might be more accurately phrased as: "…requires the assumptions/models of the core's evolving sedimentation rate between dated intervals."

 Yes we agree, the sentence will be revised to reflect the above changes.

Line 90:

I think this is a but unfair to Bchron: instead of 'resulting in extreme sedimentation rate variability', it simply posits the full range of possibility wherever there are no prior constraints on sedimentation rates.   This is arguably pretty sensible, and it represents a useful counter point to methods that assume a priori knowledge of sedimentation rates.

 We can rephrase these sentences to read "Minimal sedimentation rate constraints often result in age models with larger uncertainties than other software packages…"

Line 109:

Again on the sedimentation rate prior issue: does a prior on sedimentation rate not 'beg the question' with regard to down-core changes in age, requiring simply a single point to be anchored in time?  This seems like a very (overly) strong constraint to apply, does it not?

I think the confusion here is that the prior describes sedimentation rate *variability*, i.e., it allows for simulation of changes in sedimentation rate that are primarily used to estimate age uncertainties *between* radiocarbon dates or other absolute age estimates. (A prior that specifies a low level of sed rate variability might also reduce the fit to absolute ages if fitting those ages requires large, rapid sed rate variability; thus, the apparent sed rate changes might be ascribed to 14C dating uncertainty.)

In Undatable, the rate at which uncertainty grows between 14C dates is based on a user-specified parameter which is not based on a statistical analysis of sedimentation rate variability in any set of cores. Although the set of cores we use is not comprehensive and future work might improve our prior,

the approach of BIGMACS to use a physically based formal prior to describe how age uncertainty varies between radiocarbon dates (and beyond the first and last date) is fundamentally different from that of Undatable.

Line 138:

Is it worth noting perhaps that this shifts the problem of assuming 'instant ocean mixing' to one of a priori knowledge of past ocean hydrography and circulation?

We discuss one possible strategy of assessing homogeneity by using LGM and modern water mass geometry reconstructions in section 4.

Table 1:

note that the 14C dates for MD99-2334K are reported only by Skinner et al., G-cubed 2003 (Skinner & Elderfield 2003 does not exist, and was omitted from the references for this reason no doubt); Skinner and Shackleton 2004, and Skinner et al., Paleoc. & Paleoclim. 2021.

Thank you for catching this, we will correct the citations here.

Figure 6: What is the reason for choosing this sediment core in particular? MD99-2334K is included in the present study, has various alternative stratigraphic age-models (aligned to the Greenland ice core event stratigraphy, and the Hulu speleothem record), as well as a reasonable 14C chronology, and a well resolved benthic d18O record. Would this not be an optimal target for testing the method? A comparison with MD95-2042 could also be made, since both also have 'alignable' planktic d18O records. Furthermore, these two cores were obtained using different coring devices resulting in very different 'apparent sedimentation rates' (due to compaction in the Kasten core and stretching in the Calypso corer), providing a useful basis for assessing the algorithm's sedimentation rate prior.

This is a good suggestion, both MD99-2334K and MD95-2042 would serve as a good example here. We chose GIK13289-2 because it is not included in the DNEA stack (the alignment target) and thus the agreement between the d18O-only and C14-only age models validate our assumption of homogeneity. If we chose MD99-2334K or MD95-2042, we would expect the d18O-only and C14-only age models to agree fairly well because both of these cores contributed to the alignment target.

Line 537: again, I would propose that it might be more transparent to refer to 'radiocarbon-refined/guided d18O alignments, or similar. I wonder what the authors think.

This sentence factually lists the data and prior (sed rate variability model) that the software uses to generate probabilistic age models. It sounds like the main concern here is again the use of the word "multiproxy," which we explained above.

The simultaneous interaction of absolute age information and relative age information, particularly during stacking, is not fully captured by "radiocarbon-refined/guided d18O alignments" because 14C ages simultaneously affect both the individual alignments of the cores in the stack (thus the stack's features) and the stack's age model. "Multiproxy" is intended to convey the integration of multiple sources of (different) age information. If there is any remaining ambiguity in our usage of the word

"multiproxy", we hope for continued correspondence in an effort to make our work as clear and accurate as possible.

I look forward to reading your views on these, and most importantly the reviewers', comments.

Sincerely

Luke Skinner

---

## Author Response (AR1)

**AC1**

We thank Reviewer 1 for the constructive and thorough comments, especially regarding the implementation of the open-source software and the detailed software review. A point-by-point response to each comment is detailed below with revisions and line numbers included from the manuscript and supplement (blue).

This paper presents an interesting evolution of the the previous d18O alignment and stacking functions. The main advancements are the the use of an empirically derived sedimentation rate prior and incorporation additional age information including radiocarbon dates and tephras, tie points, etc. While I think this will be a fine contribution to the field, I think the manuscript needs to be further clarified. The bulk of the background focuses on d18O alignment. The radiocarbon descriptions can be somewhat abbreviated and overly simplistic.

The main issue I have is that the information from which the prior was obtained is completely absent. The reader knows nothing about the sediment cores, their locations, age ranges, and depositional environments. An empirically derived prior that replaces tunable parameters is only an advancement if it is appropriate to the readers sediment core. Statements around Line 220 seem to indicate that the prior is a poor match for the data, and the prior, not the radiocarbon dates are the greatest influence on the age model in the radiocarbon-dated interval. Hence, I recommend the discussion of the prior be greatly enhanced.

Thank you for bringing this to our attention. The prior presented here is derived from the same compilation of cores as Lin et al., (2014), i.e., HMM-Match. We recognize the importance of transparency, we have included a thorough description of these cores, criteria for inclusion, and the methods of construction in section S1. In addition, we have added a section in the discussion (6.1.1) that describes the potential strengths and weaknesses, the appropriateness of this data to be used as a prior for any ocean sediment core, and the option of users to substitute alternate priors. Also, our statement on line 220 was unclear and has been misinterpreted; see below.

I suggest the authors take a close look at spellings of acronyms and names. I noticed several different (and incorrect) spellings of Obrochta, as well as some acronyms that were transposed. I've noted the former in the below line-by-line comments.

Thank you, we have fixed this.

Finally, the manuscript needs to state the system requirements for running the software. I note that it uses a parallel for loop, which is not included in the standard matlab distribution. So users without the Parallel Computing Toolbox cannot use this. Also of course the sampling of PDFs will require the stats toolbox. In my experience, most people have the stats toolbox but fewer can run a parfor loop.

While the system requirements are currently stated in the User's Manual, we have added the required tool boxes to the manuscript on line 683. "BIGMACS requires the statistics and machine learning toolbox as well as the parallel computing toolbox."

\*\*\*

Line by line comments

Lines 9 - 11: "...designed to use either age proxies (e.g., radiocarbon or tephra layers) or stratigraphic alignment (e.g., of benthic δ18O) and cannot combine age inferences from both techniques."

This is a bit misleading because other Bayesian models can indeed use oxygen isotope information -- just not in the way that is done in the present paper. A manually identified oxygen isotope tie point to a reference series can be used together with radiocarbon or tephra, so I suggest this first sentence be reworded.

We have revised this sentence to distinguish between a probabilistic alignment and tie point identification/interpolation (line 10 – 13). "Previously developed software packages that generate probabilistic age models for ocean sediment cores are designed to either interpolate between different age proxies at discrete depths (e.g., radiocarbon, tephra layers, or tie points) or perform a probabilistic stratigraphic alignment to a dated target (e.g., of benthic $\delta^{18}$O) and cannot combine age inferences from both techniques."

Lines 28 - 30: "However, this method is restricted to the last 50 ka BP, suffers from variable surface reservoir ages ..., and is often low resolution causing the age model to be highly dependent on assumptions regarding sediment accumulation rate variability."

This is a somewhat outdated viewpoint. An increasing number of labs have installed AMSs, advances in automation have reduced preparation time and cost, and the ability to reliably measure radiocarbon on trace amounts of samples is making it increasingly possible to perform high resolution radiocarbon dating. Also, technically, the current marine calibration curve now extends to 55 ka, not 50 (though I agree that radiocarbon dating at those extreme ages is problematic, and resolution does not match oxygen isotope data). One study that comes to mind is:

Ishiwa, T. et al. Temporal variation in radiocarbon pathways caused by sea-level and tidal changes in the Bonaparte Gulf, northwestern Australia. Quaternary Science Reviews 266, 107079 (2021).

We have revised the sentence to reflect the 55 ka BP end age of Marine20. In addition, we have revised the text to simply state that radiocarbon is generally lower resolution than benthic d18O (line 30-33). "However, this method is restricted to the last 55 ka BP, suffers from variable surface reservoir ages (Waelbroeck et al., 2001; Sikes et al., 2016; Stern & Lisiecki, 2013; Skinner et al., 2019), and radiocarbon data are often lower resolution than benthic $\delta^{18}$O data."

Lines 36 - 37: "Software packages exist to produce probabilistic age models using radiocarbon ages (Blaauw & Christen 2011; Lougheed & Obrachta, 2019), but none of these probabilistically combine age inferences from both dating techniques."

This description should be improved to clarify that the authors are referring to software packages that automatically find the optimal alignment to a reference series. The current descriptions reads as if none of the radiocarbon-centric models can use oxygen isotope tie points.

The sentence has been changed to clarify that it refers to continuous probabilistic alignment rather than tie point identification (line 42). "Software packages exist to produce age models by interpolating between age proxies (such as radiocarbon ages, tephra layers, or/and tie points; Blaauw & Christen 2011; Lougheed & Obrochta, 2019), or by performing a probabilistic benthic $\delta^{18}$O alignment (in which residuals between input and target records are minimized; Lin et al., 2014; Ahn et al., 2017), but none of these packages can probabilistically combine age inferences from both dating techniques."

Line 46: Beyond 55 ka

This has been changed from 50 to 55 ka BP (line 59).

Lines 48 - 49: "Sedimentation rates are realistically constrained with an empirically derived prior model rather than subjective parameter settings."

I wonder how appropriate this prior is for the possible range of sediment cores that users will inevitably throw at your model? Is it possible to specify your own priors?

It is possible to specify your own prior, which we have specified in section 6.1.1 of the revised text on line 574. "If users find that the default transition model does not allow enough sedimentation rate variability to fit the age proxies for a particular set of cores, it is also possible to use your own prior distribution (see the User's Manual). However, we have not encountered such problems in testing the software, and we encourage users to exercise caution when changing this distribution."

However, alternate priors should be based on observational data or physically realistic models of the sedimentation process. While some users won't have the interest or specialized knowledge to do this, researchers interested in these processes could publish new prior models that can be used in the software, and we have plans to develop improved priors based on additional data in the future. Another worthwhile feature to implement is the ability to have a prior that depends on the sedimentation environment. Such a prior is conceptually possible with BIGMACS. For now, the manuscript has also been revised to provide more information on the prior we use here from Lin et al., (2014) in section S1. We have provided core locations and resolutions, and more thoroughly described the method of construction in section S1. Section 6.1.1 also provides a discussion of the strengths and weaknesses of the prior.

Lines 65 - 66: "Radiocarbon ages must be calibrated from 14C years to calendar years with a calibration curve that accounts for changes in past atmospheric 14C production rates (Reimer et al., 2020; Heaton et al., 2020)"

This is a very simplified statement. Changes in the carbon cycle is also taken into account, and quite a lot of work has gone into better understanding changes in marine reservoir age for Marine20.

This sentence has been revised to reflect the variations in the magnetic field of the Sun and Earth, changes in the carbon cycle, and solar storms (line 96). "Radiocarbon ages must be calibrated from $^{14}$C

years to calendar years with a calibration curve that accounts for the changing magnetic fields of the Sun and Earth, solar storms, and variations in the terrestrial carbon cycle (Reimer et al., 2020; Heaton et al., 2020; Heaton et al., 2021)."

Lines 67 - 68: "The uncertainty of the calibrated age is a combination of the calibration curve uncertainty, the radiocarbon measurement uncertainty, and the marine reservoir age uncertainty."

To this list should be added 1) local reservoir age offset from the global mean, Delta R, which also has its own uncertainty and is 2) temporally variable (e.g., older water at downstream upwelling sites following AMOC slowdown, etc.)

This has been added (line 98). "The uncertainty of the calibrated age is a combination of the calibration curve uncertainty, the radiocarbon measurement uncertainty, the time-dependent local reservoir age offset from the calibration curve ($\Delta R$) and the associated reservoir age uncertainty."

Line 76: "LGS" should be "LSG". Also perhaps Heaton 2020 should also be cited here because Marine20 includes the BICYCLE LGS-OGCM. (see above comment starting Line 65 -- Both IntCal20 and particularly Marine20 are much more sophisticated (complicated?) than just correcting for changes in production rate.

Heaton et al., 2020 has been cited here and LGS has been corrected to LSG (line 111). "Previous studies have used different methods to estimate past reservoir ages, including using modern measurements from the Global Ocean Data Analysis Project (GLODAP, Key et al., 2004, Waelbroeck et al., 2019) and the Calib database (Reimer & Reimer, 2001), comparing stratigraphically aligned age models with radiocarbon age models (Stern & Lisiecki, 2013; Skinner et al., 2021), and modelled reservoir ages from a Large Scale Geostrophic Ocean General Circulation Model (LSG-OGCM, Butzin et al., 2020; Butzin et al., 2017, Langner & Mulitza 2019; Heaton et al., 2020)."

Lines 106 and 108: "trial and error"

As with the other models described, Undatable also comes with suggestions regarding parameter selection. Both of these sentences would probably be better without "trial and error". If it took hours to converge, then that would be "trial and error", but since it takes seconds, It's more like adjusting music volume to one's desired level through instantly received feedback, which is not "trial and error". I'd suggest rewriting as:

"Its quick runtime encourages parameter tuning, based on the authors' recommendations"

and

"These parameters have large effects on the resulting age model requiring the user to decide on the most appropriate values rather than using a prior model of sedimentation rate variability."

And I also suggest that the tunable parameters in the other models be similarly discussed. As it is, this description reads as is undatable is the only one with parameters that can be tuned. This somewhat undersells what the authors are presenting here: a model without tunable parameters.

"Trial and error" has been removed from the description of Undatable and the above changes recommended by the reviewer have been made (line 142). "Undatable (Lougheed & Obrochta, 2019) uses a Monte Carlo sampling algorithm designed to emulate statistical models of sedimentation rate variability with the goal of producing quick runtimes. Users set two parameters: a scaling parameter that scales age uncertainties at the midpoints between radiocarbon ages and a bootstrapping percent that provides a framework to address outlier radiocarbon ages. These parameters have large effects on the resulting age model, requiring the user to select appropriate values, e.g., according to recommendations in Lougheed & Obrochta, (2019), rather than relying on a prior model of sedimentation rate variability."

Line 112: "which often correlates with salinity."

yes it might loosely correspond to salinity but it's really surface evaporation - precipitation prior to deepwater formation (since I assume the author's mainly considering benthic oxygen isotopes.

Yes our main consideration is benthic d18O which does often correlate with salinity.

Line 115: "The most conservative technique for aligning records to a target is to assume that large, easily identifiable features in the signals, such as glacial terminations, occurred simultaneously, create tie points between these features, and linearly interpolate between the tie points"

There absolutely is a lag between "upstream" sites in the North Atlantic and "downstream" sites since it can take on the order of 1000 years or more for the signal to propagate with the flow of deepwater.

Yes, we agree that lags exist between benthic d18O records in many cores, and we are not trying to encourage readers to always assume that benthic d18O is synchronous. In fact, in section 4.1, we specifically advise users to align only nearby cores from within the same water mass to reduce the risk of aligning diachronous signals. Here we are trying to give background and context to benthic d18O stratigraphic alignment.

Line 135: I suggest this be better presented with the information starting Line 115.

We state this at the end of the section because it applies to each alignment method. Both tie-point identification and probabilistic alignment suffer from diachronous benthic d18O signals.

Line 165: "termed the likelihoods" remove "the"

We have remove the "the" (line 218).

Line 171 - 175 The sedimentation model is called a prior distribution which is in turn called a transition model. Perhaps this can be made more clear.

We have reworded this sentence (line 223). "The prior model represents our *a priori* understanding of sedimentation rate variability and is termed the transition model."

Line 180: "confidence" should be "credible"?

Yes. We have changed "confidence" to "credible" throughout the manuscript and provided an explanation upon the first mention of "credible intervals" in section 3.1 (line 239). "In Bayesian statistics, the parameter of interest (in this case the age of sediment at a given depth) is represented by the posterior distribution, rather than a single value. Therefore, a Bayesian 95% credible interval spans 95% of the central portion of the posterior distribution. This is compared to a frequentist 95% confidence interval, which posits that there is a 95% chance that the limits are correct and encapsulate the true value. Here the 95% credible intervals and the median age model are defined by the distribution of Monte Carlo samples drawn from the posterior distribution."

Line 190: When are the locations of these 37 cores going to be disclosed?

We have added a description of these cores in section S1 including a map of core locations.

Line 191: "However, where the previous study interpolated sedimentation rates every 1 kyr, we interpolate by 1 cm"

What is the range of sedimentation rates in the 37 cores? Is 1 cm sampling typically equivalent to a 1 ky sampling, or is the interpolation interval vastly different than that used by Lin et al?

We have provided a data description and construction method for the prior in section S1.

Line 201 - 202: "Expansion specifies a below average sedimentation rate and refers to a stretching of the local portion of the record."

This is a bit confusing as stated and doesn't become clear until the next sentence where the authors stake that "contraction ... requires squeezing" Maybe rewrite as:

"Expansion refers to a below average sedimentation necessitation stretching the local portion of the record"

Ok we have changed this (line 267). "Expansion specifies a below average sedimentation rate which effectively stretches the local portion of the record. Contraction specifies a higher sedimentation rate than the average, which requires "squeezing" the record during alignment to the target."

Lines 203 - 204: "If the local sedimentation rate is within 8% of the core's average, the state is classified as steady."

How was 8% selected? Please further clarify as is done on lines 209 - 210 regarding the 15 cm interval.

We have described the transition model and the construction method in section S1. We have maintained the same bounds as Lin et al., (2014). This study defined these states to improve computational efficiency, and 8% was selected to estimate states in which sedimentation rates remain fairly constant.

Line 220: "improves agreement between the core age models and the radiocarbon observations"

I don't understand this sentence. The age model should be based on the radiocarbon observation in the radiocarbon-dated intervals. Does this indicate that the prior is often vastly different from the data, and without changing the alpha and beta parameters relative to the previous Bayesian models, the age model obtained by BIGMACS is inconsistent with the radiocarbon dates?

The likelihood for radiocarbon ages is a student's-t distribution which scales the standard deviation of the radiocarbon measurement. The degree of scaling depends on the values of alpha and beta. Using alpha and beta values of 10 and 11 applies a smaller scaling and a more peaked student's-t distribution that is more similar to the shape of a normal distribution. Thus, age model samples have a higher probability to pass closer to the mean of the radiocarbon measurement compared to using alpha and beta values of 3 and 4. We have changed the wording of this section to explain the difference more elaborately between using values of 3 and 4 from Christen & Perez (2011), vs. using values of 10 and 11 (line 283). "While Christen & Peréz (2009) and Blaauw & Christen (2011) set the fixed parameters of ⍺ and ⍴ to three and four, we choose values of ten and eleven which produces a distribution that is more peaked and more similar to a Gaussian distribution. In other words, our student's t-distribution has smaller tails than the distribution from Christen & Perez, (2009) causing age model samples to pass closer to the mean radiocarbon age. This effectively improves agreement between the age model and the radiocarbon observations."

Lines 234 - 237: "Specifying the model as a uniform distribution will force the age model to pass through the given uncertainty range and should be used when the user is confident about the age information. Specifying a Gaussian distribution will allow the age model to pass farther from the additional age constraint."

This seems backwards to me. If I specify a tephra age as a gaussian distribution with some mean and standard deviation, the highest probability is at the mean, so the model should pass closest to the mean. But if I specify a uniform distribution, the model has an equal probability of passing anywhere. So wouldn't the user want to specify a gaussian when there is good confidence in the age constraint? Perhaps I'm not following what the authors mean to say. Is it that when there is confidence in the *other* age data, with less confidence in the specified tephra/tie point, that the authors are suggesting to use a uniform distribution? I think this statements needs to be clarified.

This is a good point. A comparison between the closeness of an age model with additional age modeled as a Gaussian vs. a Uniform is difficult to generalize for every scenario. We have revised this statement to reflect this (line 304). "Specifying the model as a uniform distribution will assign an equal probability for the age model to pass anywhere through the given uncertainty range. A Gaussian distribution will assign higher probabilities to age model samples that pass close to the mean of the additional age but allows for potentially larger residuals due to the tails of the distribution assigning non-zero probabilities."

Lines 301 - 302: "these cores contain a relatively large number of δ18O outliers (Figure 1)."

Not an appropriate text location to reference fig 1. Please add lat and lon to fig 1.

We have adjusted the citation to figure 1 and have added lat & long to the map.

Table 1: confirm the longitudes

We have specified °E and provided two decimal places.

Figure 2: a color bar for the panels A and B would be helpful.

Ok we will add a color bar to this figure.

Line 347: "... crosses in Figures 4A and 5A ..."

Figure 3 has yet to be mentioned. Confirm figure numberings. I think this should be Fig 3A and 4A. Generally Figures are numbered in the order they are mentioned in the text.

Thank you for catching this mistake. We have corrected the figure references here.

Line 356: "Figure 6 compares the DNEA and ITWA stacks"

Change to Fig 5.

We have fixed this, thank you for catching it.

Line 370: "The Gaussian process regression also creates smoother stacks than previous binning methods"

It would be very useful to the reader to see a comparison of the previous stacking methods. It would also be very helpful to add a figure showing each sediment core's d18O record plotted in a separate panel above the BIGMACS stack. This will let the reader better visualize the the smoothing due to the increased autocorrelation. This would also support the assertion on Line 385 of homogenous signals.

Comparison between the DNEA and ITWA stacks constructed with BIGMACS and the DNA and INA stack constructed with Match from Lisiecki & Stern (2016) is presented in Figure S4. We have added a reference to that figure here (line 461). "Figure S4 compares the new DNEA and ITWA stacks with the Deep North Atlantic (DNA) and Intermediate North Atlantic (INA) regional stacks from Lisiecki & Stern (2016)."

While core-specific benthic d18O records are plotted in Figures 3a and 4a, supplemental figures have been added for each core that more clearly plot individual benthic d18O records in section S3.

Line 449: "6.1.1 Radiocarbon and multiproxy age models"

Missing from the discussion of applicability is, of course, if the goal is to compare phasing between d18O records, then the multiproxy age model cannot be used and only 14C, tephras, etc. can be used.

This is work we are currently investigating. We have developed a method to calculate lags between benthic d18O records by comparing 14C-only and d18O-only age models. However, the focus of this section is the advantages of BIGMACS over other age modeling software packages.

Lines 456 - 458: "Because BIGMACS applies a prior model based on observed sedimentation rate variability (Lin et al., 2014), the age uncertainty between 14C observations returned by BIGMACS is physically realistic and less subjective than using tuned parameters in other software packages."

At this point, we still know nothing about the cores from which this prior was obtained. Where are they located? What are their water depths? What are their age ranges? Do they span glacial/interglacial terminations? While this methods does not require parameter selection, it is assuming that the prior is reasonable for the *user's* sediment cores. This is an extremely important point, and I think the authors should spend some time to demonstrate to the reader that the prior is actually appropriate. In short, I'd like to have it explained to me very clearly why the prior assumed here is both appropriate and better than selecting parameters. The statement I mentioned earlier on Line 220 gives the impression that the prior is overly informative and inconsistent with the data.

We have provided a description of the cores included in the prior, the criteria for inclusion, and the methods to calculate the sedimentation rates in section S1. In addition, we have provided a new section in the discussion that addresses the application of this prior on a global scale. In addition, we have added section 6.1.1 which discusses the advantages and disadvantages of this prior. We have specified that improvement of the prior is currently being investigated. Future versions of the software package will likely have users make a choice between multiple priors based on the sedimentation environment of their core.

Line 470: "widely space"

"spaced"

Thank you we have made this correction.

Lines 478 - 479: "an assessment of a core's absolute age uncertainty should incorporate both the absolute age uncertainty of the target/stack and alignment uncertainty."

I would suggest adding an optional age error column for the stacking target, then fold that error into the alignment uncertainty. You could output both age uncertainty obtained from that of the alignment target, in addition to the alignment uncertainty already returned. The could be added to get a total uncertainty.

This is a potential future revision of the software package, and beyond the scope of the current manuscript. BIGMACS improves upon other age modeling software packages by combining direct dating techniques with probabilistic benthic d18O alignment to construct age models and stacks. During a d18O-only alignment, the uncertainty in the resulting age model only reflects the alignment uncertainty and does not include uncertainty from the target's age model. Combining these uncertainties is an obvious next step. The process of calculating absolute age uncertainty differs for stack construction and benthic d18O alignment and would increase run times. For example, absolute age uncertainty could be calculated for alignments by generating sample alignment targets consistent with the stack/target and re-running the alignment process for each sample target. The computational cost of this calculation is such that all/most currently available automated d18O alignment software (e.g., HMM-Match) does not include this feature. We agree this would be immensely useful and hope to add it to a future version of BIGMACS.

Line 544: "Example multiproxy regional stacks"

The age models are "multiproxy" but the stacks are not.

Thank you we have changed this (line 675).

Line 547: "standard deviations include the effects of spatial variability, age uncertainty ..."

I really think that there should be an easier way for users to include the age uncertainty in the alignment target.

This is definitely something we can address in future iterations of BIGMACS.

Author contributions: It appears that the first two listed authors contributed equally. As such they should be listed as "contributed equally" somewhere around where the corresponding authors are noted. If the other authors only contributed funding for this study, then technically they should not be authors and should be acknowledged.

Yes, the first two authors contributed and equally. We have added this to the corresponding authors list. All other authors directly contributed greatly to this work to and should be listed as co-authors. There contributions are specified accordingly under 'Author Contributions' at the end of the manuscript.
* * *
I didn't do a full code review but I do have some suggestions as the authors suggest that BIGMACS is resource intensive and slow. There are several things that I see that could be optimized. While I feel that the time and memory savings on the things I am point out will be minimal, it makes me wonder if there are similar inefficiencies in the most critical parts of code.

getInitialTarget.m

Line 66, load calibration curve

Why not just load only the curves that are needed? There are much more efficient ways to read in the data. The fastest is just remove the headerline of each calcurve and use simply load(path). Small things like this, if they occur throughout the code base, can add up to a savings in runtime. Also note that "path" is a command to Get/set search the path. I'd suggest changing the variable name to "Path" or "pth".

The reason why we did not remove the header line is because we hope to make the software more user-friendly: without it, a user might be confused in understanding what each column means. However, your note regarding 'path' is reasonable: we will reflect it to the next version of the software.

Why do all this:

*tic*

*path = 'Defaults/Calibration_Curves/IntCal20.txt';*

*fileID = fopen(path);*

*CAL = textscan(fileID,'%s %s %s %s %s');*

*fclose(fileID);*

*cal_curve{1} = zeros(length(CAL{1})-1,5);*

*for k = 1:5*

   *cal_curve{1}(:,k) = str2double(CAL{k}(2:end));*

*end*

*toc*

Elapsed time is 0.390675 seconds.

when you can do simply this, which is simpler and an order of magnitude faster. Are there similar chunks of inefficient code that are resulting in slow runtime?

*tic*

*Path = 'Defaults/Calibration_Curves/IntCal20.txt';*

*fileID = fopen(Path);*

*CAL = textscan(fileID,'%d %d %d %d %d','headerlines',1);*

*toc*

Elapsed time is 0.018334 seconds.

getData.m

If you can figure out what the final size of e.g., "d18O_depth" will be, you can preallocate a matrix for better memory management and faster runtime.

Though loading files is not the bottleneck in cases that we have computed, your comment regarding an efficient coding make sense. We will reflect them to the next version of the software. However, the main limit for runtime efficiency is the use of Matlab. We do not currently have the resources to convert the MATLAB codes into Python or C++, but we hope to do this in the future.

initializeAlignment.m getAlignment.m

This function uses a parfor loop, which requires the parallel computing toolbox that not everyone will have. It will also take time to start up the parallel pool if not already running. Could check for the existence of the toolbox and if it's not installed, use for instead. If there is not a significant improvement in speed, giving the time to start up the pool, it might be better to use just a for loop.

It is true that we have presumed that users may have the parallel computing toolbox and we have stated the program requirements under code availability in the manuscript.

We thank Timothy Heaton for his statistical insights and questions regarding the sampling techniques and the application of a Gaussian Process over benthic d18O data. We provide a point-by-point response to each comment below as well as the revisions we made to the manuscript.

**Overview**
This is an interesting and well-presented paper which I enjoyed reading. I would like to thank the authors for their work. They provide a statistically-rigorous approach to combine information between various sediment cores when all these cores provide observations of the same (fairly smooth) underlying function. This is known as *stacking* the records.

I presume that the model builds on earlier work (called HMM-Stack, Ahn *et al.,* 2017) in its HMM aspect for each sediment core. At its heart, the method assumes that each core $j$ records the same underlying function, providing paired observations $(y_{ij}, d_{ij})$ where: $y_{ij} = f(\theta_j(d_{ij})) + \epsilon_{ij}$

Here $\theta_j(d_{ij})$ represents the age-depth relationship in each core (which can be based upon radiocarbon dating or any another technique). The methods uses MCMC to iterate between updating the age-depth models $\theta_j(\cdot)$ for each core; and the shared function $f(\cdot)$. Within this MCMC, $f(\cdot)$ is modelled as a Gaussian Process (GP), and the age-depth model in a more complex manner (presumably based upon an approach laid out in HMM-Stack). The method does some initial particle filtering but then seems to actually ditch that approach (using it only for initialisation) to use a Metropolis-Hastings As such the particle filtering appears somewhat redundant and could therefore be de-emphasized.

Overall, the paper is nicely written with sufficient technical detail to allow reproduction. The authors also give useful practical examples for $\delta18O$ reconstructions from several marine cores. The method is potentially adaptable to a considerable range of scenarios and will provide a significant contribution to the community (although my expectation/experience is that for records which are not as smooth, or as shared, as $\delta18O$ some bespoke modifications might be required to get the model to fit – which the authors also state).

**Statistical Comments (mainly regarding the SI):**
1) My main statistical comment is that, as a new reader, I do not sufficiently understand where the specific three state HMM age-depth model comes from. I am presuming this specific age-depth model builds on previous work. In the model, there are considered to be three states for a core. Given a particular state then the sedimentation rate follows a mixture of three log-normals restricted to being within a certain range.

This particular sedimentation model seems extremely specific, yet its justification is not really provided in either the main paper or S1. It is not clear, to a new reader, where this model comes from: either in terms of three states (with seemingly arbitrary sedimentation rate bounds) or the mixture of three log-normals within its permitted ranges (are these fixed or also estimated). What is the benefit of such a three state model, why were the boundaries chosen, and how are the parameters for each state selected? Is it somewhat arbitrary or is there geoscientific insight as to why there are three distinct states with these values?

I presume this model, and its explanation, comes from the earlier HMM-Stack work of Ahn et. al (2017). If so that is fine – it does not need to be re-justified here in detail. However I feel there does need to be an intuitive lay-person explanation in the Intro about how it builds on this earlier work and what is specifically new here. Currently the HMM model appears somewhat out of nowhere. Also S1 needs much stronger referencing to that work (to clarify a reader should look there for the justification.

The transition model here uses the same data from the transition model in HMM-Match (Lin et al., 2014). The boundaries of the three states (expansion, steady, and contraction) are also the same. We have provided a more thorough description of this data and the methods used to construct the transition model in section S1. We have also provided a section in the discussion (6.1.1) elaborating on the justification of using this prior on a global compilation of cores. Future versions of BIGMACS will likely include multiple priors for the users to choose from based on the sedimentation environment of their core site. We have stated that this is an area for future development in section 6.1.1.

2) I do not quite understand how the particle filtering is used to initialise the MCMC. How do you choose which particular particles to use (after you have run the particle filtering step) as the initialisation of your later MCMC? Do you run the MCMC many times with lots of initial starting points? How have you checked actual MCMC convergence and ensured you have explored the space fully from your initialisation?

We run the MCMC many times with lots of different initial starting points sampled by the particle filtering – in fact, particle smoother is a more appropriate terminology because we rely on not only the forward algorithm but also the backward sampling. For example, if we want to sample 1000 age paths, then we first run the particle smoother to get 1000 age paths and then feed them into the MCMC. It is hard to ensure the actual MCMC convergence theoretically, but we have tuned the number of samples and iterations from simulations with real data.

3) Outlier model – I may have misunderstood but, formally, it seems you have chosen $g()$ to depend upon $\mu$. If so, I think you probably cannot entirely ignore those observations classed as outliers in the MCMC updating. When you update the GP $\mu|O,Y,A$ I would presume that the outliers will still inform as they come from a distribution that depends upon the parameter you wish to update. Consequently, I'm not sure that formally you can ignore all the values with $O=1$ and just fit a GP to the others.

This is unlikely to make much difference in practice **so I am not saying that you need to change it** (but you should perhaps mention this is an approximation). Perhaps you could get around this by keep the same mean for the outliers as the stack but just altering/increasing the variance for the outlier component $g()$. If you do this then one would presumably still include the observations in updating the GP stack but the outliers would have less weight.

We have designed the outlier model so that it is affected by the mean and variance of the inliers, which are estimated in the stack construction step. To deal with it, we do not rely on only one sampled age paths in updating the stack. For example, we sample 100 age paths in the alignment step and outliers are sampled for each age path. To do so, we can deal with "ambiguous" d18O observations – if the chance of being an outlier is 50-50, then it is regarded as an outlier in about half of the age paths and not in the rest of the paths. After it, we do the regression on each age paths after discarding outliers and then merge those regression models into one model, which is the (updated) stack.

4) The section on length/complexity does not really tell me anything practically useful, e.g. the DNEA stack has a run time of 1.8 hours. That's partially interesting, but how many MH iterations actually is that (bearing in mind you have ditched the particle filtering by that point)? You could presumably make it arbitrarily faster/slower entirely dependent on how many iterations you run everything (optimisation, particle filters, MCMC, …). Please tell me how many MCMC iterations you performed.

Time complexities are only for allowing readers to make a rough estimate of the running time, based on the number of sediment cores, their lengths, etc. The default numbers regarding the alignment and stack construction algorithm, including the number of MCMC iterations we performed, can be found in section S6 and the User's Manual which can also be found on the GitHub page.

**More General Applied/Presentation Comments:**
1) I think it would be worth mentioning how your work links with/alongside broader *errors-in-variables* regression analysis. Fundamentally, that is rather analogous to what you are doing here – if the primary interest is in the stack rather than the age-depth model of each core which it seems to be. In errors-in-variables analysis, one has a series of observations y where $yi=f(\theta)+ \epsilon i$ but you do not know $\theta i$ precisely (you only observe $Ti= \theta i+ \eta i$). This is effectively your situation - where your sediment cores provide a specific type/structure of calendar age uncertainty $\eta i$ and is some cases the $Ti$'s are not observed at all.

In a geoscience setting, using Bayesian techniques similar to you, this is basically what we do to make the IntCal curves (e.g. Heaton et al. 2020) but there is also quite a lot of general statistical methodological literature (e.g. Bayesian approach of Cook and Stefanski, 1994) on the topic. Additionally, there is quite a lot of literature on *registration* in functional data analysis which could briefly be mentioned (e.g. book by J Ramsay and B Silverman).
I also did some work with a similar (but identical) goal – aiming to sharing age information between records using tie-points and a GP – in Heaton *et al.* (2013). This was used to create calendar ages for the Pakistan and Iberian Margin (Bard et al. 2013) , and Cariaco Basin (Hughen & Heaton 2020) data which then went into IntCal13 and IntCal20. This work was somewhat different in that we only tried to transfer dating information from one record to another and only used the tie-point ages. However it does provide a previous context where tie-points are used in a method that aims for statistical rigour rather than eye-balled tuning (with uncertainties on the contemporaneity of the ties rather like your model). Your work is however more in depth and generalisable than ours (we needed fairly simple age-depth models with multivariate covariances so owe could input then into the main IntCal curve creation) Suggest that all this only needs a brief line or two in the Intro – just to add more detail/context about how your work fits within the wider statistical research literature.

Good suggestion. We have included "errors-in-variables" regression in the introduction on line 75. "Similar to "errors-in-variables" regression, which is used to construct the Intcal20 curve due to uncertainty in both the radiocarbon measurements and their calendar ages (Reimer et al., 2020; Heaton et al., 2020), BIGMACS calculates a time series of mean and variance for benthic $\delta^{18}$O by performing Gaussian process regressions (Rasmussen and Williams, 2006) across MCMC age model samples."

We have also cited and described Heaton et al., 2013 in section 2.2 on line 172. "Heaton et al., (2013) presents an age model construction method which uses a Gaussian process regression to interpolate between benthic $\delta^{18}$O tie points. The method incorporates uncertainty from the target age model, tie point identification, and interpolation between tie points and was used to construct chronologies for records incorporated into the IntCal13 curve (Reimer et al., 2013). Heaton et al., (2013) argue against using a deterministic automated alignment process (e.g., Lisiecki and Lisiecki, 2002) due to a lack of uncertainty estimates and concerns about aligning across different proxy types which may differ in sensitivity to climate responses. We assert that using BIGMACS to align across a set of sediment cores with homogeneous signals of the same proxy (such as benthic $\delta^{18}$O in neighbouring cores), addresses these concerns. BIGMACS formally incorporates multiple sources of age uncertainty to create probabilistic alignments that are both more informative and less subjective than tie point identification."

2) My colleagues (when I tried to suggest a similar approach to them to map all features across records for other proxies) were very cautious. They felt that, for many records, the entirety of the proxy could not be mapped between cores. They rather believed that, for many proxies, it was often only the sharp/main transitions that were shared between records and they did not want to match everything.

I feel this point, that users must consider if trying to match every feature is something that will work for their proxy/data, should be made very explicitly. You do mention this in the manuscript but it is somewhat hidden and only appears towards the end (in the middle of the section on Strengths/Weaknesses on lines 520-525). I feel this caveat needs to be made considerably more prominent in the Intro/Conclusion when discussing GPs so readers will not misunderstand.
I am not a sufficient expert here, but it may be that benthic $\delta 18O$ is more globally homogeneous than many other proxies (and the method must be used with considerable caution for some other proxies where responses can be antithetic).

While benthic d18O has been traditionally used as a proxy for global ice volume, studies have identified offsets during T1 between different locations. These offsets have been attributed to changes in circulation rates, asynchronous surface signals, and water mass boundary changes. In section 4.1 we outline a strategy to identify cores that have likely been bathed by similar water masses and, thus, likely share a homogeneous benthic d18O signal.

3) Your Marine sites are very spread out and will not be expected to have the same regional offset $\Delta R$ from one another. You have chosen a mean of $\Delta R$=0 for all the cores but then quite a large uncertainty ($\sigma$= 200) on $\Delta R$ to account for uncertainty. Again this is probably fine, as you have chosen a fasirly large value (and I think everything will be somewhat led by the fitting of the many $\delta 18O$ measurements anyway). However, I would suggest that you might advise users to initialize a different $\Delta R$ for each core using the Reimer and Reimer (2017) database.

We do not advise people to choose $\Delta R$=0 if they have other information available. The belief is that, at least during the Holocene, any regional $\Delta R$ will remain roughly constant over time and so will be applicable along the core (as regional upwelling/ocean depth might remain relatively constant). If you choose an independent $\Delta R$ from one observation to the next then you do not model dependence in
**Note: This is a fairly minor point that I doubt will affect your results due to the volume of $\delta 18O$ observations. If it is a lot of work (or the marine core sites you use do not have $\Delta R$ estimates) then I suggest you just add a caveat/explanation for the paper (rather than redo everything).**

Yes, we could use the delta-R package in conjunction with an independent dating technique. Stratigraphically aligning a surface proxy to NGRIP offers a potential method to independently date cores at the Iberian Margin. For example, Skinner et al., (2019) aligned the abundance of N. pachyderma to calculate a time dependent reservoir age.  Future iterations of BIGMACS may support alignment and stacking of planktonic d18O in which case the delta-R package and BIGMACS could produce reservoir age uncertainties that not only include the 14C measurement and calibration error but also the uncertainty from the probabilistic alignment. However, calculating a reservoir age for the cores in the ITWA stack is more challenging, as far as we know there is not an established target for these cores. This is an exciting direction to pursue, but we feel it is outside the scope of this work. We have added a caveat mentioning the Calib database on line 435. "To calibrate radiocarbon ages to calendar years, we use the Marine20 calibration curve (Heaton et al., 2020), a constant reservoir age offset (ΔR) equal to zero, and a reservoir age standard deviation of 200 years (although it should be noted that future users

can find potential reservoir age offsets using the Calib database; Reimer & Reimer, 2001). We make no corrections for the different planktonic species used to measure radiocarbon in each core (see Table 1 for data citations)."

4) Is there a reason as to why the sedimentation rates of Lin et al. (2014) are applicable elsewhere? This seems like a considerable assumption. Hence while it is potentially a strength of your method to provide automated selections of sedimentation rates it is also a considerable danger if other use it as a black box when it is not appropriate. At the very least, you must ensure that any user inputs their data on the same measurement scale (i.e. m or cm) as the analysis you did for Lin et al. (2014).

Yes, we agree. We have described the data and methods used to construct the transition model in section S1 and we have included a new discussion that addresses the strengths and weaknesses of the transition model in section 6.1.1. We assert that, because the prior is constructed from a *global* compilation of cores spanning a large depth range, it is an appropriate prior for most ocean sediment cores. However, this is a current area we are working to improve, and future iterations of BIGMACS will include updated (and perhaps regionally specific) priors.

**Smaller Specific Points:**
**Main Document:**
1) Line 66 – it is not only 14C production rate changes which cause variations in past atmospheric 14C/12C levels but also rearrangements of the carbon cycle (see e.g. Heaton *et al.* 2021). Suggest minor rewording to acknowledge this.

This has been revised to reflect the processes described in Heaton et al., 2021 on line 96. "Radiocarbon ages must be calibrated from $^{14}$C years to calendar years with a calibration curve that accounts for the changing magnetic fields of the Sun and Earth, solar storms, and variations in the terrestrial carbon cycle (Reimer et al., 2020; Heaton et al., 2020; Heaton et al., 2021)."

2) Line 370 are your stack estimates smoother because you use a GP which is fundamentally a significant smoother? Or due to other factors such as averaging over calendar ages? Also does the smoothed version lose genuine features - are the features you say you smooth/lose thought to be genuine phenomena?

The stacks constructed with BIGMACS are smooth because the Gaussian process regression is a continuous algorithm and thus does not have discrete steps that previous methods employed. Also, the degree of smoothing is determined by the choice of kernel. In addition, stack samples are drawn over the entire range of age model samples for each core (if there is not a clear feature to align there will be larger age uncertainty and potentially a larger stack standard deviation). Furthermore, because stacking is inherently a smoothing process by filtering out features that do not exist in every core (i.e., noise). Figure S4 compares the DNEA and ITWA stacks with the DNA and INA stacks published in Lisiecki & Stern (2016). We see that timing and magnitude of each MIS stage agrees in the DNEA and DNA stacks. Data composing the ITWA stack is inherently noisier, and it is difficult to decipher which features are noise (perhaps caused by sediment disturbances) and which are climate signals. However, BIGMACS handles this data appropriately by increasing the standard deviation of the stack. We have added a citation to Figure S4 on line 461. "Figure S4 compares the new DNEA and ITWA stacks with the Deep North Atlantic (DNA) and Intermediate North Atlantic (INA) regional stacks from Lisiecki & Stern (2016)."

3) Figure 3 and Figure 4 – in the panel As showing the final stack, can you overlay the posterior mean estimate on top of the observations (rather than underneath where currently it can't be seen)

We have made this change, and we have changed the circles for individual d18O data points to stars to further increase the visibility of the stack.

4) Line 473-474 - *Users should be aware that the age uncertainties returned by BIGMACS for age models generated by multiproxy alignment or stacking do not include the age uncertainty of the alignment target.* I do not understand this comment about an alignment target – based upon your SI you suggest you can use your method on records where there is no a priori alignment target (i.e. when you just have a selection of cores each with their own 14C dates). Have I misunderstood?

Yes, BIGMACS requires an initial alignment target for the first iteration of stack construction. After the first iteration, the alignment target becomes the stack that was drawn in the last iteration. This sentence is referring to the age uncertainty resulting from an alignment. The uncertainty returned by BIGMACS is an alignment uncertainty, and shows the spread of possible alignments. However, every alignment target will also have age uncertainty, and this uncertainty is not included in the alignment process. A future version of BIGMACS may include this, the absolute age uncertainty of a benthic d18O aligned age model should reflect both the alignment uncertainty and the targets age uncertainty. We have added an explanation of the stacking procedure in the introduction on line 70.

"Another functionality of BIGMACS is the automated construction of multiproxy benthic $\delta^{18}$O stacks using an iterative process that simultaneously considers the probabilistic fit to both absolute age information (e.g., from radiocarbon dates) and relative age information from alignment of all cores' benthic $\delta^{18}$O signals. Age models for each core are constructed by aligning benthic $\delta^{18}$O to the stack from the previous iteration, and then a new stack is calculated from the aligned $\delta^{18}$O from every core. Radiocarbon ages (if included) help constrain the age models for their respective cores during each iteration of stack construction. Similar to "errors-in-variables" regression, which is used to construct the Intcal20 curve due to uncertainty in both the radiocarbon measurements and their calendar ages (Reimer et al., 2020; Heaton et al., 2020), BIGMACS calculates a time series of mean and variance for benthic $\delta^{18}$O by performing Gaussian process regressions (Rasmussen and Williams, 2006) across MCMC age model samples. The resulting stack variance is a combination of both age model uncertainty from individual cores and the spread of benthic $\delta^{18}$O from every core. This method requires fewer cores than previous stacking methods (e.g., Ahn et al., 2017; Lisiecki & Stern, 2016) and, thus, allows users to construct target stacks from a small number of neighbouring cores that share homogeneous $\delta^{18}$O signals. "

We have also added explanations of absolute uncertainty and relative uncertainty at multiple places in the manuscript.

Line 66: "The distribution of MCMC samples at a given depth of a radiocarbon age model reflects the absolute age uncertainty of the sediment. However, $\delta^{18}$O age model uncertainty reflects only the relative age uncertainty and excludes the absolute age uncertainty of the alignment target."

Line 511: "Furthermore, the uncertainty for the $\delta^{18}$O-only age model reflects only the *alignment* uncertainty. The absolute age uncertainty would be a combination of the alignment uncertainty and the absolute age uncertainty from the DNEA stack."

**Suppl. Information**

1. There are repeated uses of sigma to mean many things – unclear what the values that are updated in S4 refer to. Equally what are the h's – need to be made somewhat clearer?

Because the model is complicated and the number of alphabets are limited, we could not avoid using some Greek alphabets multiple times with variations including additional bars and/or subscripts. We have added a description of h's at the beginning of S4.

2. More detail is needed on the parameter choices for the age-depth model – can refer to other work if this is suitable.

We are not sure which section, age-depth model, or parameter choices this comment is referring to?

3. Minor point – the likelihoods are not probabilities (the densities are continuous)

Thank you we have fixed this.

4. S5 – There is some referencing to other sections that has gone wrong: "The stack construction algorithm first iterates steps in subsections S4.2, S4.3 and S4.4 until convergence and then update the new one by the method in S4.1."

Thank you for catching this, we have corrected it.

There is no S4.4. Also, do you mean S5.1 at the end rather than S4.1?

Yes thank you, we have corrected the numbering of supplementary sections.

**General Questions (as I'm interested – not requiring further work):**

1) I tried work on a similar topic a few years ago. I found that the lack of homogeneity in the underlying functions we considered (and that it was only some features that were shared) made the method hard to implement in practice. I didn't get it to work very well (hence it remains unpublished).

Do you think that there is something special about the $\delta 18O$ signals you use that mean the features are highly shared between cores? Do you expect it to work as well for more challenging/variable functions/proxies? Do you think there is a danger that you get into highly multi-modal fits in some cases which the MCMC will not fully explore – or is your age-depth prior sufficiently strong to avoid that?

Because benthic temperature changes are relatively homogenous compared to SST changes we expect major features (such as termination events and other MIS stages) to be shared between signals from different cores. However, we do stress the importance of selecting cores based on their water mass history. It is more difficult to claim that a surface proxy, such as a planktonic d18O, is homogeneous. Yet the Western Pacific Warm Pool provides one location that may have experienced homogeneous SST changes across late Pleistocene glacial cycles. We are currently investigating this and working to construct a planktonic d18O stack.

Age models produced with BIGMACS can be multi-modal if the direct age estimates (radiocarbon or additional ages) occur at a lower frequency than the cycle of the aligned signal. For example, we have observed multi-modal distributions in the 40 kyr world when using additional ages derived from magnetic reversals.

2) How much of a difference do the 14C measurements really make a difference when you have to match 2000 $\delta 18O$ observations? Do these swamp the independent calendar age information? Might there be use in having a dependency in the proxy measurements you wish to construct (from one observation to the next)?

We compare age uncertainty produced from C14-only age models, d18O-only age models, and multiproxy age models in Figure S3. For the age models in this study the multiproxy age models have the smallest uncertainty, followed by C14-only and finally d18O-only. Furthermore we find that the multiproxy age models agree well with the C14-only age models (panel b of Figures 3 and 4). Disagreement between multiproxy, d18O, and C14 age models for a single core could be a result of C14 age errors, heterogeneous d18O signals, or/and sediment disturbances. The extent to which BIGMACS will favor one proxy over the other depends on the total calibrated radiocarbon age uncertainty, the standard deviation of the target stack, and the likelihoods for C14 and d18O data. Both likelihoods are modeled as a student's t-distribution, however the distribution applied to C14 data has smaller tails than d18O data. It is difficult to generalize the degree to which different resolutions control the age model construction process.

3) As a statistician, I think it is a bit of a shame that all of the material on the methods itself has been moved to the SI. I appreciate I am biased and that many readers will be much more interested in the results than technical details.

We have found it difficult to find a balance between simplification and statistical rigor. Our goal is to create a manuscript that targets the broader palaeoceanographic community.

**References**
Ahn, S., Khider, D., Lisiecki, L. E., and Lawrence, C. E.: A probabilistic Pliocene–Pleistocene stack of benthic δ18O using a profile hidden Markov model, 2, https://doi.org/10.1093/climsys/dzx002, 2017.
Bard E., Ménot G., Rostek F., Licari L., Böning P., Edwards R.L., Cheng H., Wang Y., Heaton T.J., 2013. Radiocarbon calibration/comparison records based on marine sediments from the Pakistan and Iberian margins. Radiocarbon 55,1999-2019.
Heaton, T., Bard, E., Hughen, K., 2013. Elastic Tie-Pointing—Transferring Chronologies between Records via a Gaussian Process. *Radiocarbon, 55*(4), 1975-1997. Doi:10.2458/azu_js_rc.55.17777
Heaton T.J., Blaauw M., Blackwell P.G., Ramsey C.B., Reimer P.J., Scott E.M., 2020. The IntCal20 approach to radiocarbon calibration curve construction: a new methodology using Bayesian splines and errors-in-variables. Radiocarbon 62,821-63.
Heaton T.J., Bard E., Ramsey C.B., Butzin M., Köhler P., Muscheler R., Reimer P.J., Wacker L., 2021. Radiocarbon: A key tracer for studying Earth's dynamo, climate system, carbon cycle, and Sun. Science 374: eabd7096.
Hughen K.A., Heaton T.J., 2020. Updated Cariaco Basin 14C calibration dataset from 0–60 cal kyr BP. Radiocarbon 62,1001-43.
Reimer, R. W. and Reimer, P. J.: An Online Application for ΔR Calculation, Radiocarbon, 59, 1623–1627, https://doi.org/DOI: 10.1017/RDC.2016.117, 2017.

Cook, J. R. and Stefanski, L. A. (1994). Simulation-Extrapolation Estimation in Parametric Measurement Error Models. Journal of the American Statistical Association, 89(428):1314{1328.

**AC3**

Now that a sufficient number of reviews have been submitted, I would like to invite you provide a point-by-point response to each of the review comments. I would like to underline the importance of addressing a key issue that has been raised in the reviews, namely: the 'empirically constrained' sedimentation rate prior that is applied in the matching algorithm. One issue is that the validity and applicability of this prior, across a range of sedimentary contexts, does not appear to have been fully assessed in a transparent manner in the manuscript – and indeed seems doubtful.

Below I add a few remarks of my own, in case these are helpful for considering what revisions you would undertake, and I invite your response to these too. I find your study of particular interest, and I hope that my comments will be seen as useful.

We thank the editor Luke Skinner for his summaries of the reviews and insights into radiocarbon and benthic d18O age model construction. We believe the revised manuscript addresses all concerns raised by the reviewers. Below is a point-by-point response to the editor's comments.

Title/general ethos:

In general, I think it might be useful to more clearly delineate the distinction between alignment and 'dating' at the core of the manuscript (even though the difference between relative and 'absolute/numerical' ages is indeed noted in the paper a few times). Creating a benthic d18O stack is one thing, aligning to a benthic d18O stack is another, and dating a sediment core is yet another.

Thank you for your comment here, it is clear that we needed further clarification on this point. Your delineation of three separate processes: alignment, stacking, and dating has been true in the past; however, our new approach integrates these three steps together, such that this is no longer the case. Dating from absolute age proxies is now incorporated simultaneously with the determination of relative ages from alignment. This is why we feel it is appropriate to call these "multiproxy age models;" it is not simply the assignment of a radiocarbon age model after alignment or stacking. We have better described the nuance and novelty of how our approach combines age model construction with stack construction in the second to last paragraph of the introduction beginning on line 70.

"Another functionality of BIGMACS is the automated construction of multiproxy benthic $\delta^{18}O$ stacks using an iterative process that simultaneously considers the probabilistic fit to both absolute age information (e.g., from radiocarbon dates) and relative age information from alignment of all cores' benthic $\delta^{18}O$ signals. Age models for each core are constructed by aligning benthic $\delta^{18}O$ to the stack from the previous iteration, and then a new stack is calculated from the aligned $\delta^{18}O$ from every core. Radiocarbon ages (if included) help constrain the age models for their respective cores during each iteration of stack construction. Similar to "errors-in-variables" regression, which is used to construct the Intcal20 curve due to uncertainty in both the radiocarbon measurements and their calendar ages (Reimer et al., 2020; Heaton et al., 2020), BIGMACS calculates a time series of mean and variance for benthic $\delta^{18}O$ by performing Gaussian process regressions (Rasmussen and Williams, 2006) across MCMC age model samples. The resulting stack variance is a combination of both age model uncertainty from individual cores and the spread of benthic $\delta^{18}O$ from every core. This method requires fewer cores than

previous stacking methods (e.g., Ahn et al., 2017; Lisiecki & Stern, 2016) and, thus, allows users to construct target stacks from a small number of neighbouring cores that share homogeneous $\delta^{18}O$ signals."

The only way that benthic alignment provides age constraints is if one proposes to have prior knowledge of how local/regional deep-water T and d18Osw relate to insolation, e.g. based on a hypothesis for how insolation paces ice volume, and how changes in ice volume are linked to deep water T changes and/or influence deep ocean d18Osw at a given location in the ocean. The latter sequence of hypotheses can give age constraints that are of ~millennial accuracy at best. In such a context, radiocarbon dates (even with ~centennial uncertainties in reservoir age offsets) obviously can provide a refinement.

Thank you for bringing this to our attention, we also believe readers will need more clarification. BIGMACS is not attempting to use the benthic d18O as an indicator of absolute time (e.g., via orbital tuning) as you describe here. d18O is used only to constrain relative ages between cores (or a target stack). We have added more clarification of this point in the introduction on line 66. "The distribution of MCMC samples at a given depth of a radiocarbon age model reflects the absolute age uncertainty of the sediment. However, $\delta^{18}O$ age model uncertainty reflects only the relative age uncertainty and excludes the absolute age uncertainty of the alignment target. BIGMACS does not use any orbital tuning unless users choose to align to a target stack that has been orbitally tuned."

We have also added a description in the section 5 on line 508. "While the radiocarbon and multiproxy age models have direct age constraints via radiocarbon ages, the $\delta^{18}O$-only age model provides only relative age constraints. Furthermore, the uncertainty for the $\delta^{18}O$-only age model reflects only the *alignment* uncertainty. The absolute age uncertainty would be a combination of the alignment uncertainty and the absolute age uncertainty from the DNEA stack."

Our motivation in developing this software is the same point you make above, there is added value in simultaneously considering benthic d18O and 14C during the alignment process.

The inverse is unlikely to be true: age constraints on benthic d18O are unlikely to be precise enough, even to constrain changes in radiocarbon reservoir age offsets of order 100-1000 years.

We agree with this point and do not intend to make such a claim in the manuscript. We have added an explicit clarification that orbital tuning is not used by BIGMACS on line 68. "BIGMACS does not use any orbital tuning unless users choose to align to a target stack that has been orbitally tuned."

Alternatively, if the core notion of the manuscript and algorithm is the simple transferral of a radiocarbon chronology (or the pooling of radiocarbon dates) between sediment cores via a stratigraphic alignment of benthic d18O, then again it is not quite a case of 'combining age constraints from radiocarbon and benthic d18O'. Rather, it is one of radiocarbon dating of a stratigraphic alignment/stack.

The main concern here appears to be a nuance in how the age model technique is described. Your description of a "transferal of a radiocarbon chronology (or the pooling of radiocarbon dates) between sediment cores via a stratigraphic alignment of benthic d18O" is a reasonable summary of the software during stack construction. However, BIGMACS can also include other types of age constraints, offers an alignment-only mode which does not require any 14C dates, and also constrains age models based on a

prior for sedimentation rate variability. While the concept is simple, the probabilistic model leads to rigorous statistical results that simultaneously combines many pieces of information.

What the software does is more complex than "radiocarbon dating of an alignment/stack" because the alignment/stacking is performed simultaneously with consideration of the 14C age information. Because the software integrates relative age information from benthic d18O simultaneously with 14C age constraints, we feel that "multiproxy" is an appropriate description of the age modeling method. In fact, we show in Figure S3 that considering both d18O alignment and 14C dates (ie, multiple proxies) in age model construction produces smaller confidence intervals than either alone (in large part due to the pooling of 14C dates from multiple cores, which requires alignments of the cores). We have attempted to further clarify this with further explanation in the introduction with a description of stack construction beginning on line 65.

As such, my own feeling is that the manuscript might more accurately be framed in terms of 'refining orbitally-tuned benthic d18O age models using radiocarbon constraints', e.g. in the title and through the text.

There seems to be some misunderstanding here. No orbitally tuned ages are incorporated in BIGMACS unless a user chooses to align to an orbitally tuned target. None of our examples use any age information derived from orbital tuning. If radiocarbon dates are used in stack construction, the final age model produced will derive almost entirely from 14C over the time period for which 14C dates are available (because the stack is iteratively updated to agree with absolute age constraints). We have clarified that BIGMACS does not use orbital tuning on line 68. "BIGMACS does not use any orbital tuning, however users can choose to align to a target stack that has been orbitally tuned."

In a similar vein, it seems to me that describing the age models as 'multi-proxy' is a little misleading: my own expectation was initially of something like that described in line 522. I would again suggest that the process tackled in the present study be described as something like 'radiocarbon-refined single proxy alignment'.

Thank you for bringing this to our attention. You are likely not the only reader who may initially misinterpret our use of the word multiproxy, and we have attempted to clarify this in the introduction on line 59. "We use the term "multiproxy" to indicate the combined inference from two types of "age proxies": absolute age information provided by radiocarbon and relative age information from the stratigraphic alignment of benthic $\delta^{18}O$. Note that this method is distinct from an alignment of multiple climate proxies (e.g., benthic and planktonic $\delta^{18}O$)."

We have also included further emphasis in the methods section on line 233. "The posterior distribution of a multiproxy age model includes likelihoods returned by the radiocarbon emission model, the benthic $\delta^{18}O$ emission model, and the additional age emission model."

Although currently BIGMACS can only align benthic d18O data, we hope to develop a future version will be equipped to handle alignments of multiple climate/sediment proxies.

We refer to the age models as multiproxy because information from both radiocarbon ages and benthic d18O-alignment are simultaneously integrated during age model construction. If the user has an input core with radiocarbon data, benthic d18O data, and an independently dated alignment target that they believe has a synchronous benthic d18O signal, then the combination of both proxies determine the age

uncertainty of the age model. Figure S3 demonstrates that the multiproxy age models have smaller uncertainties than their single proxy counterparts for every core in this study.

BIGMACS can also incorporate "multiproxy" (relative and absolute) age information on timescales beyond the limit of 14C dating with the use of additional sources of absolute age information (called "additional ages" in the software). For example, tephra layers or tie points to speleothems can provide additional absolute age information to improve an age model, compared to a "single proxy" alignment based solely on benthic d18O alignment.

Although other age modeling software can combine absolute age information from 14C with non-14C sources, no other dating software combines these absolute age estimates with automated signal alignment (ie, probabilistically derived relative ages). For example, Undatable only incorporates relative age information if the user specifies an absolute age estimate for a discrete tie point. This is dramatically different from the continuous probabilistic relative (aligned) age estimates that BIGMACS integrates with absolute age constraints.

Perhaps the source of concern with the term "multiproxy" is that benthic d18O and 14C are proxies for different things. 14C is a proxy for absolute age whereas benthic d18O is a proxy for climate/seawater properties and, thus, relative age. We have added more explanation to clarify our intended meaning of "multiproxy," but we assert that our use of the term is both technically correct and appropriately conveys the integration of multiple dating techniques. However, if a decision about publication of this manuscript ultimately rests on our inclusion of the word multiproxy in its description, we are willing to discuss alternatives.

Line 25:

This line is not quite correct: the accuracy with which ocean sediment cores can reconstruct the *timing* of past climate events, depends on.. the.. age model. The accuracy of proxies is a separate (thorny) matter.

We have revised this sentence on line 27 to state "The accuracy with which ocean sediment core data can reconstruct the timing of past climate events depends on the quality of the core's age model (i.e., estimates of age as a function of core depth)."

Line 48:

"Sedimentation rates are realistically constrained…."

As pointed out by the first Reviewer, it seems we must take this on faith, whereas there is burden of demonstration here.

In our revision we have included an in-depth description of the data and construction methods of the transition model in section S1. In addition, we have added a section in the discussion (6.1.1) that summarizes the strengths and weaknesses of this prior and plans for future improvements.

Line 65:

In general, there is a need to be precise when describing radiocarbon procedures. Radiocarbon dates need to be calibrated to account for past changes in the initial radiocarbon concentration of the fossil entity's 'parent reservoir' (atmosphere, surface ocean, etc.), which may change due to 14C-production changes and/or other carbon cycle processes. This crops up again on Line 72: planktonic foraminiferal radiocarbon dates must be corrected for 'reservoir age offsets' (relative to the atmosphere) only if using a record of past atmospheric radiocarbon concentration/activity for the calibration. In principle, a 'marine calibration curve' might be used instead, with different potential corrections needed as a result.

Thank you for catching this. We have revised these sentences to reflect the processes described in Heaton et al., (2021) and the potential to calibrate planktonic radiocarbon ages with the Marine20 curve.

Line 96 now states "Radiocarbon ages must be calibrated from $^{14}$C years to calendar years with a calibration curve that accounts for the changing magnetic fields of the Sun and Earth, solar storms, and variations in the terrestrial carbon cycle (Reimer et al., 2020; Heaton et al., 2020; Heaton et al., 2021)."

Line 105 has been revised as well. "Planktonic foraminiferal radiocarbon dates must be corrected for the reservoir age of the surface ocean relative to the atmosphere or calibrated with a curve that accounts for the reservoir age of the surface ocean (e.g., the Marine20 curve; Heaton et al., 2020)."

Line 79:

"…requires simulating the core's sedimentation rate."

I think this might be more accurately phrased as: "…requires the assumptions/models of the core's evolving sedimentation rate between dated intervals."

Yes, we agree. The sentence has been revised to reflect the above changes on line 113. "Constructing a sediment core age model, which estimates sediment ages for all core depths, from a sequence of radiocarbon ages requires assumptions or models of the core's evolving sedimentation rate between dated intervals."

Line 90:

I think this is a but unfair to Bchron: instead of 'resulting in extreme sedimentation rate variability', it simply posits the full range of possibility wherever there are no prior constraints on sedimentation rates. This is arguably pretty sensible, and it represents a useful counter point to methods that assume a priori knowledge of sedimentation rates.

Thank you for bringing this to our attention; we concede that our original phrasing was unclear. We have rephrased these sentences on line 126 to read "Bchron requires few user-specified parameter settings and posits less prior knowledge on sedimentation rate constraints, thus age models constructed with Bchron often have larger age uncertainties than other software packages…"

Line 109:

Again on the sedimentation rate prior issue: does a prior on sedimentation rate not 'beg the question' with regard to down-core changes in age, requiring simply a single point to be anchored in time?  This seems like a very (overly) strong constraint to apply, does it not?

I think the confusion here is that the prior describes sedimentation rate *variability*, i.e., it allows for simulation of changes in sedimentation rate that are primarily used to estimate age uncertainties *between* radiocarbon dates or other absolute age estimates. (A prior that imposes a low level of sed rate variability might also reduce the fit to radiocarbon ages if fitting those ages requires large, rapid sed rate variability; thus, the apparent sed rate changes can also be affected by 14C dating uncertainty.)

In Undatable, the rate at which uncertainty grows between 14C dates is based on a user-specified parameter which is not based on a statistical analysis of sedimentation rate variability in any set of cores. Although the set of cores we use is not comprehensive and future work might improve our prior, the approach of BIGMACS to use a physically based formal prior to describe how age uncertainty varies between radiocarbon dates (and beyond the first and last date) is fundamentally different from that of Undatable and establishes a framework that can be improved upon over time.

Line 138:

Is it worth noting perhaps that this shifts the problem of assuming 'instant ocean mixing' to one of a priori knowledge of past ocean hydrography and circulation?

Yes, thank you. We have added a sentence describing some potential causes of benthic d18O lag times on line 187. "Causes of offsets in the timing of benthic $\delta^{18}O$ change include asynchronous surface signals, changes in deep ocean water mass geometry, or/and different transit times for northern and southern sourced water masses (Gebbie, 2012)."

Table 1:

note that the 14C dates for MD99-2334K are reported only by Skinner et al., G-cubed 2003 (Skinner & Elderfield 2003 does not exist, and was omitted from the references for this reason no doubt); Skinner and Shackleton 2004, and Skinner et al., Paleoc. & Paleoclim. 2021.

Thank you for catching this. It looks like Skinner et al., 2021 cites Skinner et al., PNAS. (2014) for the radiocarbon dates for MD99-2334K as well. We have changed our citations to include Skinner et al., (2003); Skinner & Shackleton (2004); Skinner et al., (2014); Skinner et al., 2021. We for missing citations here.

Figure 6: What is the reason for choosing this sediment core in particular?  MD99-2334K is included in the present study, has various alternative stratigraphic age-models (aligned to the Greenland ice core event stratigraphy, and the Hulu speleothem record), as well as a reasonable 14C chronology, and a well resolved benthic d18O record.  Would this not be an optimal target for testing the method?  A comparison with MD95-2042 could also be made, since both also have 'alignable' planktic d18O records.  Furthermore, these two cores were obtained using different coring devices resulting in very

different 'apparent sedimentation rates' (due to compaction in the Kasten core and stretching in the Calypso corer), providing a useful basis for assessing the algorithm's sedimentation rate prior.

This is a good suggestion, both MD99-2334K and MD95-2042 would serve as a good example here. We chose GIK13289-2 because it is not included in the DNEA stack (the alignment target) and thus the agreement between the d18O-only and C14-only age models helps validate our assumption of homogeneity. If we chose MD99-2334K or MD95-2042, we would expect the d18O-only and C14-only age models to agree fairly well because both of these cores contributed to the alignment target.

Line 537: again, I would propose that it might be more transparent to refer to 'radiocarbon-refined/guided d18O alignments, or similar. I wonder what the authors think.

This sentence factually lists the data and prior (sed rate variability model) that the software uses to generate probabilistic age models. Perhaps the main concern here is again the use of the word "multiproxy"?

The simultaneous interaction of absolute age information and relative age information, particularly during stacking, is not fully captured by "radiocarbon-refined/guided d18O alignments" because 14C ages simultaneously affect both the individual alignments of the cores in the stack (thus the stack's features) and the stack's age model. "Multiproxy" is intended to convey the integration of multiple types of age information (absolute age from 14C and relative ages from d18O). If there is any remaining ambiguity in our usage of the word "multiproxy", we welcome additional feedback on how we can make the text as clear and accurate as possible.

I look forward to reading your views on these, and most importantly the reviewers', comments.

Sincerely

Luke Skinner

---

## Author Response (AR2)

We want to express our gratitude to Anonymous Reviewer 1, Timothy Heaton (Reviewer 2), and Luke Skinner (the Editor) for their valuable feedback and time invested in reviewing our manuscript. Based on their comments, we have made revisions to improve the description of the data used to construct the prior and we have moved this description from section S1 and section 3.2. Specifically we hope to have clarified that the prior includes data from 37 ocean sediment cores after the quality control. The cores are listed in table S1 and their locations are displayed in Figure 1.

Thank you,
Devin Rand